# META-UCF: UNIFIED TASK-CONDITIONED LoRA GENERATION FOR CONTINUAL LEARNING IN LARGE LANGUAGE MODELS

**Shilin Xiao**[1,5†]**, Tianxiang Xu**[2†]**, Canran Xiao**[3*]**, Weihao Luo**[4]**, Liwei Hou**[5]**, Chuangxin Zhao**[6]

[1]Nanjing University of Finance and Economics, [2]Peking University
[3]Shenzhen Campus of Sun Yat-sen University, [4]College of Science and Technology, Ningbo University, [5]Hunan Airon Tech, [6]Institute of Automation, Chinese Academy of Sciences
`2120233029@stu.nufe.edu.cn; xiaocr3@mail.sysu.edu.cn`

## ABSTRACT

Large language models are increasingly deployed in settings where new tasks arrive continuously, yet existing parameter-efficient finetuning (PEFT) methods either bloat linearly with the task horizon or sacrifice deep adaptation, leaving catastrophic forgetting unresolved. We aim to achieve *memory-constant, on-the-fly* adaptation for a frozen LLM facing an unbounded stream of tasks. To this end we propose Meta-Unified Contrastive Finetuning (META-UCF), which encodes each task into a lightweight layer-normalised mean embedding and feeds it to a single hypernetwork that instantly generates rank-$r$ LoRA updates for every transformer layer; a meta-contrastive coupled with orthogonality objective further steers task embeddings into near-orthogonal directions, preserving past knowledge without inner-loop gradients. On four benchmark streams—Std-CL 5, Seq-GLUE 7, Long-CL 15 and TRACE-8—Meta-UCF raises average accuracy by up to 2.2 pp and cuts forgetting by 13 % relative to the strongest LoRA baseline, while using the parameters of a single adapter. By decoupling continual learning from parameter growth, Meta-UCF provides a practical path toward scalable, low-resource lifelong language modelling.

## 1 INTRODUCTION

Large language models (LLMs) underpin modern NLP systems yet remain costly to personalise for the continually growing set of downstream tasks demanded by real applications (Yao et al., 2023; Zhu et al., 2024; Kamath et al., 2024; Zhang et al., 2025b; Feng et al., 2026; Zhang et al., 2025a). Updating a multi-billion-parameter backbone after each task is prohibitive in compute, storage and energy (Ding & Shi, 2024; Jegham et al., 2025); nevertheless, accurate and rapid adaptation *without forgetting* previous skills is crucial for life-long AI agents deployed at scale (Fawi, 2024; Liao et al., 2024; Jiang et al., 2024).

Recent parameter-efficient finetuning (PEFT) techniques—adapters, prefixes and, most notably, LoRA—shrink per-task overhead from full weights to a few percent (Hu et al., 2022; Houlsby et al., 2019; Li & Liang, 2021; Zhang et al., 2024). However, when tasks arrive sequentially existing variants allocate one static slot per task (Wang et al., 2023; Tiwari et al.,

Figure 1: Existing approaches keep adding a separate adapter for every new task. Meta-UCF instead trains *one shared hypernetwork* that, from a task embedding, generates the required low-rank update on the fly—eliminating linear parameter growth.

---

*Corresponding author
†These authors contributed equally to this work

2025; Yang et al., 2025), so model size still grows linearly with the horizon and subspace scheduling becomes brittle. Prompt-retrieval methods (Wang et al., 2022; Song et al., 2023; Bohao et al., 2024) avoid weight growth but leave the backbone frozen, limiting reasoning transfer.

This work probes a deeper gap in PEFT: current methods treat each incoming task as an isolated *patch*—they either allocate a new low-rank slot or attach a prompt, leaving the backbone untouched—yet offer no mechanism to *re-organise* the knowledge already stored as the task stream grows. Consequently, model size expands linearly, and task interference is addressed post-hoc with orthogonality heuristics (Wang et al., 2023; Tiwari et al., 2025).

We close this gap by reframing sequential PEFT as a *generative* problem, and propose Meta–Unified Contrastive Fine-Tuning (META-UCF). Our key idea is to encode every task into a compact *layer-normalised mean* vector and feed it to a single hypernetwork that *generates* rank-$r$ LoRA updates for *all* transformer layers on the fly(Figure 1). A meta-contrastive objective pushes task embeddings towards near-orthogonality, while a lightweight orthogonality penalty prevents their generated directions from collapsing. Thus a frozen LLM remains both *plastic*—via instant, conditioned updates—and *stable*—because only the hypernetwork learns and its memory footprint is constant.

Our contributions are threefold: **(i)** We introduce a task-conditioned LoRA hypernetwork with an orthogonality-aware meta objective that eliminates linear parameter growth. **(ii)** We prove expressivity bounds for low-rank hypernetworks and a PAC-Bayes generalisation bound over task streams. **(iii)** Meta-UCF achieves new state-of-the-art accuracy and forgetting on four benchmarks while using a constant-sized model; ablations reveal robust accuracy–latency trade-offs.

## 2 RELATED WORK

**Parameter-efficient adaptation.** Early adapter modules (Houlsby et al., 2019) and prefix/p-tuning (Li & Liang, 2021; Liu et al., 2021) reduce finetuning cost by inserting tiny task-specific weights. LoRA pushes this idea further by applying low-rank updates directly to attention and FFN matrices (Hu et al., 2022). A recent surge of LoRA variants targets continual scenarios: O-LoRA orthogonalises task subspaces to curb interference(Wang et al., 2023), N-LoRA re-parameterises updates to avoid collision(Yang et al., 2025), while GRID (Tiwari et al., 2025) and Adaptive-SVD (Nayak et al., 2025) compress adapter banks under a shared orthonormal basis. Despite strong empirical gains, these methods allocate a *static* slot per task, leaving memory proportional to the task horizon and requiring manual scheduling of subspaces. META-UCF replaces the slot bank with a single hypernetwork that *generates* LoRA factors on demand, retaining the footprint of a *single* task regardless of stream length.

**Continual learning for language models.** Classical replay and regularisation ideas (e.g. EWC(Kirkpatrick et al., 2017), GEM(Lopez-Paz & Ranzato, 2017), LwF(Li & Hoiem, 2017)) have been ported to transformers but scale poorly when the backbone exceeds billions of parameters. Prompt-based approaches, such as ProgPrompt and L2P(Wang et al., 2022), store small textual or embedding prompts in a memory bank; ConPET (Song et al., 2023), JARe(Bohao et al., 2024) and Continual-T0(Scialom et al., 2022) couple such prompts with contrastive objectives. Yet these frameworks depend on explicit prompt retrieval at inference time and cannot modify deeper representations, limiting accuracy on reasoning-heavy streams. Meta-UCF instead learns a compact task embedding that drives low-rank updates *throughout* the network, yielding stronger plasticity while preserving frozen parameters.

## 3 METHOD

**Meta–Unified Contrastive Fine-Tuning (Meta-UCF)** tackles the *continual-learning* setting in which a stream of tasks $\{\mathcal{T}_1, \mathcal{T}_2, \dots\}$ arrives sequentially and the backbone model must adapt without retaining a separate adapter for each task. Meta-UCF equips a *frozen* LLM backbone with a *single* hyper-network that *generates* low-rank LoRA updates on the fly, conditioned on a compact *task embedding* constructed from a small replay buffer. Figure 2 illustrates the training flow: a support set drawn from replay memory is encoded by the *frozen* backbone and averaged to obtain a task embedding $e_k$; this embedding is fed to the shared hyper-network $g_{\mathbf{\Phi}}$, which instantly generates rank-$r$ LoRA factors for every transformer layer; the backbone augmented with these factors pro-

cesses the current task's query batch, and the joint task, orthogonality, contrastive, and bias losses back-propagate to update $\boldsymbol{\Phi}$ alone, leaving the backbone weights unchanged.

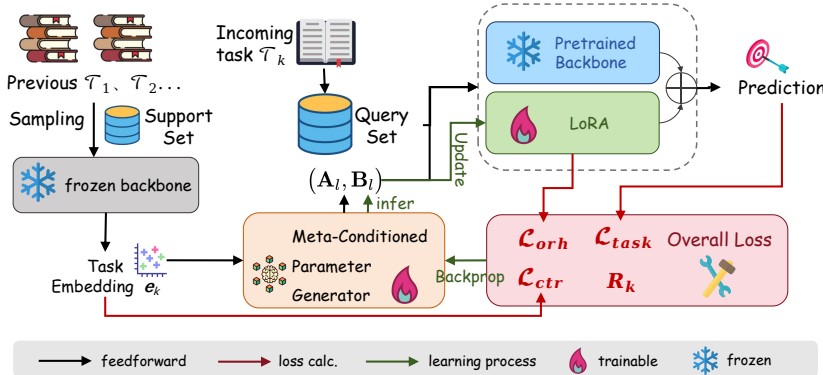

**Figure 2: Meta-UCF pipeline**: a support set from memory yields the task embedding $e_k$, the generator $g_{\boldsymbol{\Phi}}$ produces LoRA updates $(\mathbf{A}_l, \mathbf{B}_l)$ for the frozen backbone, and the current task's query batch drives losses $\{\mathcal{L}_{\text{task}}, \mathcal{L}_{\text{orth}}, \mathcal{L}_{\text{ctr}}, R_k\}$ whose gradient updates only the generator.

## 3.1 TASK EMBEDDING ACQUISITION

A task embedding $e_k$ should (i) *summarise* the latent structure of the current task $\mathcal{T}_k$, (ii) be *stable* under mini-batch sampling noise, (iii) remain *parameter-free* so that it can be computed on-the-fly at deployment time, and (iv) live in the same representation space as the backbone so that geometric notions (e.g. cosine similarity) are meaningful. Formally, let the frozen backbone be a function $f_{\boldsymbol{\Theta}_0} : \mathcal{X} \to \mathbb{R}^d$ that maps an input $x \in \mathcal{X}$ to its CLS hidden state $\mathbf{h} = f_{\boldsymbol{\Theta}_0}(x)$. Given a support set $\mathcal{S}_k = \{x_s\}_{s=1}^{S_k}$, we would like $e_k = \text{Pool}\big(\{\mathbf{h}_s\}_{s=1}^{S_k}\big)$ to satisfy

$$\text{sim}(e_i, e_j) \approx \delta_{ij} \quad \text{with} \quad \text{sim}(\mathbf{a}, \mathbf{b}) = \frac{\mathbf{a}^\top \mathbf{b}}{\|\mathbf{a}\| \|\mathbf{b}\|}, \tag{1}$$

so that task embeddings are *approximately orthogonal* across different tasks.

**Layer-normalised mean pooling.** A simple yet powerful choice is the *layer-normalised mean*:

$$e_k = \text{LN}\Big(\frac{1}{S_k}\sum_{s=1}^{S_k}\mathbf{h}_s\Big), \qquad \mathbf{h}_s = f_{\boldsymbol{\Theta}_0}(x_s), \tag{2}$$

where LN denotes layer normalisation acting on the feature dimension. Equation 2 enjoys three favourable properties:

1. *Unbiasedness.* Let $\mu_k = \mathbb{E}_{x \sim P_k}[f_{\boldsymbol{\Theta}_0}(x)]$ be the true task mean under the episode distribution $P_k$. Then $\mathbb{E}[e_k] = \text{LN}(\mu_k)$.

2. *Variance decay.* If $\text{Cov}[\mathbf{h}_s] = \Sigma_k$, then $\text{Cov}\big[\frac{1}{S_k}\sum_s \mathbf{h}_s\big] = \frac{1}{S_k}\Sigma_k$, i.e. the variance shrinks at a rate $O(S_k^{-1})$.

3. *Scale equivariance.* Layer normalisation removes arbitrary feature-wise scale, making $e_k$ insensitive to backbone re-scaling that may occur during pre-training.

equation 2 can be interpreted as the first-order term in a Fisher-kernel expansion. Writing $\ell(x; \boldsymbol{\Theta}_0)$ for the log-likelihood of $x$ under the frozen model, the Fisher score is $\mathbf{g}(x) = \nabla_{\boldsymbol{\Theta}}\ell(x; \boldsymbol{\Theta})\big|_{\boldsymbol{\Theta}=\boldsymbol{\Theta}_0}$. Under a linearisation of the backbone, $\mathbf{g}(x)$ is proportional to the hidden state $\mathbf{h}$, hence the average $\bar{\mathbf{g}}_k = \frac{1}{S_k}\sum_s \mathbf{g}(x_s)$ yields the same embedding as equation 2 up to a constant. From classical theory,

$$K(e_i, e_j) = \bar{\mathbf{g}}_i^\top \mathbf{F}^{-1} \bar{\mathbf{g}}_j, \tag{3}$$

with $\mathbf{F}$ the Fisher information matrix, is a *kernel* that measures task similarity.

**Streaming update.** During continual training the support set grows; we therefore maintain exponential-moving-average (EMA) estimates:

$$e_k^{(t)} = \text{LN}\big((1-\rho)\, e_k^{(t-1)} + \rho\, \bar{\mathbf{h}}^{(t)}\big), \quad \bar{\mathbf{h}}^{(t)} = \frac{1}{|\mathcal{B}_t|} \sum_{x \in \mathcal{B}_t} f_{\boldsymbol{\Theta}_0}(x), \tag{4}$$

with decay $\rho \in (0,1)$ and mini-batch $\mathcal{B}_t$ drawn from $\mathcal{S}_k$, yielding $O(d)$ memory overhead irrespective of $S_k$.

**Distance-preserving normalisation.** Finally, note that applying LN followed by $\ell_2$-normalisation projects all embeddings onto the unit hypersphere, so that

$$\text{sim}(e_i, e_j) = 1 - \tfrac{1}{2}\, \|e_i - e_j\|_2^2, \tag{5}$$

showing that Euclidean distance and cosine similarity coincide—a useful property for contrastive objectives.

## 3.2 META-CONDITIONED PARAMETER GENERATOR

The core challenge in continual learning is to avoid linear growth in the number of trainable parameters as new tasks arrive. Meta-UCF therefore replaces a bank of per-task adapters with a single hyper-network $g_{\boldsymbol{\Phi}}$ that synthesises LoRA updates for every Transformer layer on demand.

**Generator architecture.** Let $e_k \in \mathbb{R}^d$ be the task embedding from § 3.1. We first compute a task code $\mathbf{z}_k = \text{MLP}_{\text{task}}(e_k) \in \mathbb{R}^h$, $h < d$, using a two-layer MLP with GELU activation. For each layer index $l$ we retrieve a learned positional embedding $\mathbf{p}_l \in \mathbb{R}^h$ and concatenate: $\tilde{\mathbf{z}}_{k,l} = [\mathbf{z}_k; \mathbf{p}_l]$. Two low-rank projection heads then generate LoRA factors

$$\big(\mathbf{A}_l, \mathbf{B}_l\big) = \Big(\text{reshape}_{d \times r}\big(W_A \tilde{\mathbf{z}}_{k,l}\big), \; \text{reshape}_{r \times d}\big(W_B \tilde{\mathbf{z}}_{k,l}\big)\Big), \tag{6}$$

where $W_A, W_B \in \mathbb{R}^{dr \times 2h}$ are shared across layers.*

**LoRA injection.** Given the base weight $\mathbf{W}_l \in \mathbb{R}^{d \times d}$ of layer $l$, the generator applies a rank-$r$ update

$$\mathbf{W}_l^{(k)} = \mathbf{W}_l + \alpha\, \mathbf{B}_l\big(e_k; \boldsymbol{\Phi}\big)\, \mathbf{A}_l\big(e_k; \boldsymbol{\Phi}\big), \quad \alpha = \tfrac{1}{r}. \tag{7}$$

The scaling $\alpha$ follows LoRA convention so that the update norm remains comparable across different ranks.

**Complexity analysis.** The generator's parameters decompose as $|\boldsymbol{\Phi}| = |\text{MLP}_{\text{task}}| + 2hL + 2hdr$, yielding total computational cost $\mathcal{O}\big(|\boldsymbol{\Phi}| + Ldr\big)$ per forward pass—independent of the number of tasks $K$.

## 3.3 META OBJECTIVE

For each episodic task $\mathcal{T}_k$ we draw disjoint *support* $\mathcal{S}_k$ and *query* $\mathcal{Q}_k$ sets. All losses are evaluated on $\mathcal{Q}_k$ after a single hyper-network forward pass conditioned on $\mathcal{S}_k$. Concretely, in a single meta-training step we first sample a meta-batch of $K$ tasks and, for each task $\mathcal{T}_k$, draw disjoint support $\mathcal{S}_k$ and query $\mathcal{Q}_k$ (Yao et al., 2024). The frozen backbone encodes each support example $x \in \mathcal{S}_k$ into a CLS vector $h_s \in \mathbb{R}^d$, which is layer-normalised and averaged to produce the task embedding $e_k \in \mathbb{R}^d$ defined in §3.1. The generator then maps $e_k$ to a lower-dimensional code $z_k \in \mathbb{R}^h$, combines it with the layer embedding $p_\ell$, and outputs rank-$r$ LoRA factors $A_\ell \in \mathbb{R}^{d \times r}$ and $B_\ell \in \mathbb{R}^{r \times d}$ for every Transformer layer. These factors are injected into the backbone, which is run once on the query batch $\mathcal{Q}_k$ to obtain predictions and query CLS states; stacking the latter forms $H_k \in \mathbb{R}^{|\mathcal{Q}_k| \times d}$, on which the task loss $\mathcal{L}_{\text{task}}^{(k)}$, the orthogonality penalty $\mathcal{L}_{\text{orth}}$, and the bias regulariser $R_k$ are computed. In parallel, the set of task embeddings $\{e_k\}_{k=1}^K$ is fed to the contrastive loss $\mathcal{L}_{\text{ctr}}$ defined below. A single backward pass through this graph updates only the generator parameters $\boldsymbol{\Phi}$, while the backbone parameters and layer-normalisation statistics remain frozen.

---

*Sharing $W_A, W_B$ keeps $|\boldsymbol{\Phi}|$ sub-linear in $L$ while allowing layer-specific outputs via $\mathbf{p}_l$.

**Task Accuracy.**

$$\mathcal{L}_{\text{task}}^{(k)} = \frac{1}{|\mathcal{Q}_k|} \sum_{(x,y) \in \mathcal{Q}_k} \ell\big(f_{\boldsymbol{\Theta}_0, \boldsymbol{\Delta}(e_k)}(x), y\big). \tag{8}$$

This is the standard supervised objective that drives the generated adapters to fit the labels of each episode, providing the "plasticity" needed to acquire new tasks.

**Orthogonality Penalty.** Let $\mathbf{H}_k \in \mathbb{R}^{|\mathcal{Q}_k| \times d}$ stack each query's CLS state. Define the pair-wise Frobenius overlap

$$\Omega_{ij} := \frac{1}{|\mathcal{Q}_i| \, |\mathcal{Q}_j|} \big\| \mathbf{H}_i^\top \mathbf{H}_j \big\|_F, \tag{9}$$

and set

$$\mathcal{L}_{\text{orth}} = \sum_{i<j} \Omega_{ij}^2. \tag{10}$$

Intuitively, $\mathbf{H}_k$ collects the $d$-dimensional query representations for task $\mathcal{T}_k$, and $\Omega_{ij}$ measures how much the subspaces spanned by $\mathbf{H}_i$ and $\mathbf{H}_j$ overlap; penalising $\Omega_{ij}^2$ discourages different tasks from sharing the same dominant directions, improving stability by reducing cross-task interference in the adapted backbone.

**Meta-Contrastive Separation.** With task embeddings $\mathbf{z}_k := e_k$, the InfoNCE loss is

$$\mathcal{L}_{\text{ctr}} = -\frac{1}{K} \sum_{k=1}^{K} \log \frac{\exp\big(\text{sim}(\mathbf{z}_k, \mathbf{z}_k^+)/\tau\big)}{\sum_{j \neq k} \exp\big(\text{sim}(\mathbf{z}_k, \mathbf{z}_j)/\tau\big)}, \tag{11}$$

where $\text{sim}(\mathbf{a}, \mathbf{b}) = \mathbf{a}^\top \mathbf{b}/(\|\mathbf{a}\| \, \|\mathbf{b}\|)$, $\tau$ is a temperature, and $\mathbf{z}_k^+$ denotes the embedding of an *independent* support minibatch $\mathcal{S}_k^+$ drawn from the same task $\mathcal{T}_k$ as $\mathcal{S}_k$, computed with the same frozen backbone and layer-normalised mean pooling. In other words, $(\mathbf{z}_k, \mathbf{z}_k^+)$ form two IID "views" of the same task distribution, giving a simple task-level data augmentation without introducing extra trainable modules. Once the embeddings are $\ell_2$-normalised, maximising the InfoNCE objective over cosine similarities enforces angular separation between tasks on the unit hypersphere, a standard and numerically stable choice in meta-contrastive learning. In Meta-UCF, $\mathcal{L}_{\text{ctr}}$ shapes this *input* geometry of the generator by keeping task codes nearly orthogonal, while $\mathcal{L}_{\text{orth}}$ regularises the *output* geometry of the adapted backbone by discouraging overlap between the query subspaces $\mathbf{H}_i$ and $\mathbf{H}_j$; the two regularisers therefore operate at complementary levels to balance plasticity and stability.

**Dynamic Bias Calibration.** For a binary sensitive attribute $g \in \{0, 1\}$, the *demographic-parity gap* is

$$R_k = \Big| \mathbb{E}_{x \sim P(x \,|\, g=0, \mathcal{T}_k)} f_{\boldsymbol{\Theta}_0, \boldsymbol{\Delta}(e_k)}(x)$$
$$- \mathbb{E}_{x \sim P(x \,|\, g=1, \mathcal{T}_k)} f_{\boldsymbol{\Theta}_0, \boldsymbol{\Delta}(e_k)}(x) \Big|. \tag{12}$$

Gradients w.r.t. the generator parameters $\boldsymbol{\Phi}$ are scaled by $\sigma(-\beta R_k)$, where $\sigma$ is the sigmoid and $\beta > 0$ a sensitivity hyper-parameter.

**Overall Loss.**

$$\mathcal{L}_{\text{meta}} = \sum_{k=1}^{K} \Big( \mathcal{L}_{\text{task}}^{(k)} + \lambda_o \mathcal{L}_{\text{orth}} + \lambda_c \mathcal{L}_{\text{ctr}} + \lambda_b R_k \Big). \tag{13}$$

Thus $\mathcal{L}_{\text{meta}}$ remains a simple episodic objective: for each task, the supervised loss encourages adaptation, the orthogonality and contrastive terms regularise the geometry of task codes and representations, and the bias term gates updates based on the demographic-parity gap.

**Outer-Loop Optimisation** We employ a first-order MAML variant with zero inner-loop gradient steps. At each iteration we (a) sample a batch of tasks, (b) construct $\mathcal{S}_k, \mathcal{Q}_k$ for each, (c) compute $\mathcal{L}_{\text{meta}}$, and (d) update $\boldsymbol{\Phi}$ via AdamW. Backbone parameters $\boldsymbol{\Theta}_0$ and layer-norm statistics remain frozen.

Table 1: Overall comparison. Darker shading indicates better performance.

| Method | Std-CL 5 | Long-CL 15 | Seq-GLUE 7 | TRACE-8 |
|---|---|---|---|---|
| LoRA | 78.3 | 61.4 | 75.9 | 55.6 |
| O-LoRA | 80.1 | 63.4 | 76.8 | 57.3 |
| CL-LoRA | 82.0 | 64.3 | 78.7 | 59.7 |
| OA-Adapter | 82.7 | 65.9 | 78.9 | 59.8 |
| SAPT | 83.2 | 68.1 | 79.6 | 60.7 |
| N-LoRA | 83.5 | 68.1 | 80.2 | 61.0 |
| L2P | 80.0 | 62.0 | 75.8 | 56.0 |
| SPARC | 81.4 | 61.5 | 76.4 | 58.5 |
| InsCL | 80.1 | 63.2 | 77.6 | 58.8 |
| EWC | 78.8 | 61.6 | 74.2 | 55.4 |
| LwF | 79.5 | 61.2 | 76.8 | 56.9 |
| Replay-LoRA | 80.5 | 63.8 | 77.1 | 57.9 |
| META-UCF (r=8, All) | 85.6 | 70.7 | 82.7 | 63.4 |
| META-UCF (r=8, Top-Half) | 84.9 | 70.1 | 82.1 | 62.9 |
| META-UCF (r=4, All) | 84.3 | 69.0 | 81.6 | 62.0 |

| Method | Forgetting Ratio | | | | Backward Transfer | | | |
|---|---|---|---|---|---|---|---|---|
| | Std-5 | Long-15 | GLUE-7 | TRACE-8 | Std-5 | Long-15 | GLUE-7 | TRACE-8 |
| LoRA | 12.5 | 18.3 | 10.9 | 21.2 | −1.8 | −4.2 | −1.3 | −5.5 |
| O-LoRA | 10.4 | 16.0 | 9.8 | 19.5 | −1.2 | −3.5 | −1.0 | −4.8 |
| CL-LoRA | 9.6 | 14.9 | 8.7 | 17.1 | −0.6 | −2.5 | −0.6 | −3.8 |
| OA-Adapter | 8.7 | 14.2 | 8.3 | 17.1 | −0.7 | −2.7 | −0.6 | −3.8 |
| SAPT | 7.2 | 13.3 | 7.3 | 16.0 | −0.3 | −2.7 | −0.2 | −3.6 |
| Adaptive SVD | 7.5 | 13.0 | 7.4 | 15.9 | −0.4 | −2.2 | −0.3 | −3.3 |
| N-LoRA | 7.1 | 12.4 | 7.1 | 15.5 | −0.3 | −2.0 | −0.2 | −3.1 |
| L2P | 11.0 | 17.6 | 10.0 | 19.7 | −1.5 | −4.0 | −1.2 | −5.3 |
| SPARC | 14.4 | 19.5 | 11.7 | 21.4 | −2.6 | −3.8 | −2.2 | −4.7 |
| InsCL | 9.5 | 14.3 | 8.6 | 18.1 | −1.1 | −2.4 | −1.2 | −3.5 |
| EWC | 13.6 | 17.4 | 12.5 | 20.3 | −2.4 | −4.1 | −1.8 | −4.8 |
| LwF | 12.8 | 16.2 | 11.6 | 18.8 | −2.1 | −3.5 | −1.2 | −4.3 |
| Replay-LoRA | 9.3 | 15.0 | 8.5 | 18.2 | −1.1 | −3.1 | −0.9 | −4.3 |
| META-UCF | 6.2 | 11.5 | 6.3 | 14.2 | 0.2 | −1.5 | 0.1 | −2.5 |

(a) **Average Accuracy** (%, ↑). The two Meta-UCF variants use fewer adapted parameters or layers under comparable budgets.

(b) **Forgetting Ratio** (%, ↓) and **Backward Transfer** (BWT, ↑). Lower FR and higher BWT indicate better stability.

**Inference** During deployment, a small support set ($S \leq 16$) from a *previously unseen* task is enough to produce $e_{\text{new}}$ and hence $\Delta(e_{\text{new}})$ *without optimisation*. The frozen backbone combined with the generated adapters executes the downstream prediction, enabling *one-model-for-all-tasks* operation with negligible memory overhead.

For the complete implementation pseudocode of Meta-UCF, please refer to Algorithm 1 and Algorithm 2 in Appendix B.1.

# 4 EXPERIMENTS

## 4.1 EXPERIMENTAL SETTINGS

**Evaluation Benchmarks.** To assess the performance of Meta-UCF, we conduct experiments across four distinct task sequences, following the methodology established in continual LoRA fine-tuning: (i) *Std-CL 5*: A standard five-task sequence used for text classification, encompassing tasks from AG News, Amazon, Yelp, DBpedia, and Yahoo in that order. (ii) *Seq-GLUE 7*: A task sequence based on the GLUE benchmark, which includes the tasks CoLA, SST-2, MRPC, QQP, QNLI, RTE, and MNLI, providing a comprehensive test of natural language understanding (NLU) transfer. (iii) *Long-CL 15*: A more complex sequence of 15 tasks, extending the Std-CL 5 by incorporating tasks from the GLUE, SuperGLUE, and IMDb datasets. This benchmark is evaluated with three distinct task orderings. (iv) *TRACE-8*: A multi-domain eight-task stream covering areas like domain-specific question answering, multilingual comprehension, code completion, and mathematical reasoning. All datasets are transformed into the SEQ2SEQ instruction format as outlined by Qin et al. (2024), and comprehensive statistics are available in the Appendix.

**Evaluation Protocol.** We report the standard continual-learning metrics: *Average Accuracy* (AA), *Forgetting Ratio* (F.R.), and *Backward Transfer* (BWT). For datasets with multiple metrics (e.g. accuracy & F1) we follow GRID (Tiwari et al., 2025) and average them into a single score. All results are averaged over three random seeds.

**Baselines.** We benchmark Meta-UCF against three groups of baselines in parameter-efficient continual adaptation of large language models and NLP systems: *(i) Adapter/parameter-efficient continual learning* — CL-LoRA (He et al., 2025), TreeLoRA (Qian et al., 2025), OA-Adapter (Wan et al., 2025), N-LoRA(Yang et al., 2025), and SAPT (Zhao et al., 2024). *(ii) Prompt-based continual adaptation* — L2P (Wang et al., 2022), SPARC (Jayasuriya et al., 2025), and InsCL (Wang et al., 2024). *(iii) Memory / regularisation strategies* — Elastic Weight Consolidation (EWC) (Kirkpatrick et al., 2017), Learning without Forgetting (LwF) (Li & Hoiem, 2017), and Replay-LoRA (Pillai, 2025).

**Backbone Models.** We consider four recent 7–13B checkpoints: LLAMA-3-8B, QWEN-1.5-7B, DEEPSEEK-7B, and MISTRAL-7B. rank-$r = 8$ LoRA adapters are inserted into every qkv and MLP projection.

Table 2: Combined results. Left: Alpaca pre-tuning effects on **MMLU** and **Std-CL 5**. Right: single-factor ablations of META-UCF on **Std-CL 5** and **Long-CL 15**.

| Method | MMLU ↑ | Std-CL 5 ↑ |
|---|---|---|
| *w/o CL* | | |
| LLaMA-7B | 34.4 | — |
| Alpaca-LoRA | 37.5 | — |
| Alpaca-LoRA-CL | 23.3 | 46.7 |
| Alpaca-inc-LoRA-CL | 28.6 | 33.1 |
| Alpaca-*O*LoRA-CL | 33.6 | 76.8 |
| **Alpaca-Meta-UCF-CL** | **36.2** | **80.5** |

(a) Zero-shot **MMLU** and downstream **Std-CL 5** accuracy after Alpaca pre-tuning.

| Variant | Std-CL 5 | | Long-CL 15 | |
|---|---|---|---|---|
| | Acc. ↑ | FR ↓ | Acc. ↑ | FR ↓ |
| Full Meta-UCF | **85.6** | **6.2** | **70.7** | **11.5** |
| w/o $\mathcal{L}_{orth}$ | 83.9 | 7.8 | 68.5 | 13.2 |
| w/o $\mathcal{L}_{ctr}$ | 84.1 | 7.2 | 68.9 | 12.7 |
| w/o bias calibration | 84.6 | 6.9 | 69.4 | 12.0 |
| CLS mean → last CLS | 82.1 | 9.5 | 66.3 | 15.1 |
| static LoRA (no generator) | 80.3 | 11.1 | 64.9 | 17.0 |

(b) Single-factor ablations of META-UCF.

**Optimisation Details.** Unless noted, we train each task for a single epoch with AdamW ($\beta_{1,2} = 0.9, 0.98$), learning-rate $3\times10^{-5}$, batch 64, sequence length 512, and weight-decay 0.01. Meta-UCF regulariser weights are fixed across streams: $\lambda_o = 0.5$, $\lambda_c = 1.0$, $\lambda_b = 0.1$, bias sensitivity $\beta = 4$, EMA decay $\rho = 0.2$, and support size $S_k = 32$. All runs fit on a single NVIDIA A100 80G; Long-CL 15 uses ZeRO-2 across four GPUs to keep wall-clock under 24 h.

## 4.2 MAIN RESULTS

The results in Table 1a show that META-UCF delivers the highest average accuracy on all four streams, improving over the strongest prior baseline (N-LoRA) by +1.7 pp on STD-CL 5 and +2.2 pp on the heterogeneous TRACE-8. Table 1b further indicates that Meta-UCF not only reduces forgetting to a new low (e.g., 6.2% on STD-CL 5) but also turns backward transfer nearly neutral or mildly positive, whereas all competing methods remain negative. Together, these gains confirm that task-conditioned LoRA generation—combined with orthogonality and bias-aware meta objectives—yields both superior accuracy and markedly improved stability across short, long, and domain-diverse continual-learning streams. To ensure that the gains on heterogeneous streams are not driven by a single domain, we also compute per-task AA gaps between Meta-UCF and N-LoRA on Long-CL 15 and TRACE-8, which are reported in Appendix D.2.

To test the zero-shot performance of the Meta-UCF method, we follow the O-LoRA(Wang et al., 2023) protocol: first, we instruction-tune **LLaMA-7B** on the ALPACA dataset using rank-8 LoRA, and then perform continual training on the STD-CL 5 (order 1) stream. As Table 2a shows, META-UCF attains the highest downstream accuracy (**80.5%**), surpassing the strongest baseline Alpaca-O-LoRA-CL by +3.7 pp. Crucially, it does so *without sacrificing generalisation*: the zero-shot MMLU score rises to 36.2%, close to the single-task Alpaca-LoRA (37.5%) and considerably above all prior continual variants. These results confirm that task-conditioned LoRA generation preserves the general knowledge acquired during Alpaca pre-tuning while providing superior resistance to catastrophic forgetting on subsequent tasks.

## 4.3 ABLATION STUDY

To isolate the impact of each design component, we conduct *single-factor* ablations on the two representative streams—STD-CL 5 and LONG-CL 15. As shown in Table 2b, removing $\mathcal{L}_{orth}$ or $\mathcal{L}_{ctr}$ lowers accuracy by 1.1–1.9 pp and adds $\approx$1.5 pp forgetting, evidencing their joint role in drift control. Bias calibration is less critical but still helps, especially on longer streams. Replacing the mean-pooled embedding with a single CLS vector costs 3.1 pp on STD-CL 5, and using a fixed LoRA slot hurts both metrics most, underscoring the need for task-conditioned generation.

## 4.4 SENSITIVITY ANALYSIS

**Parameter Sensitivity** We vary every hyper-parameter that could plausibly influence META-UCF and measure average accuracy (mean $\pm$ std over three seeds) on STD-CL 5 and LONG-CL 15. Results in Figure 3 indicate that the method is remarkably robust: most settings fluctuate within $\pm$1 pp of the default, and no single factor dominates performance.

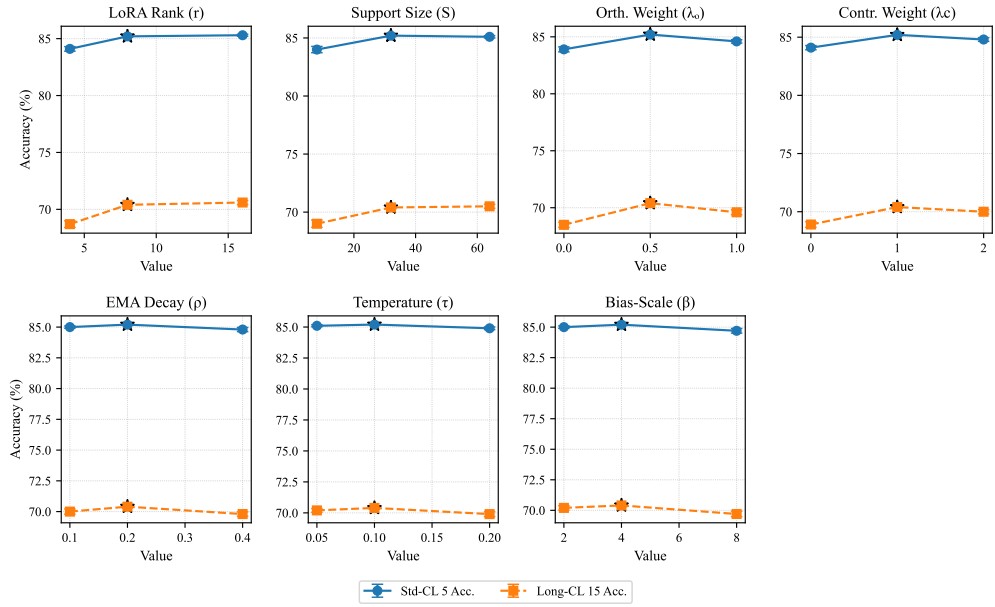

Figure 3: Sensitivity of META-UCF to key hyper-parameters.

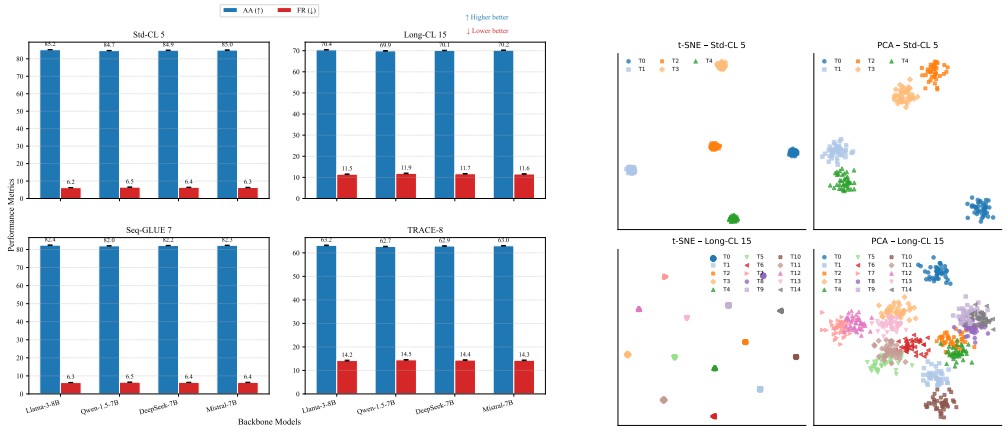

(a) Meta-UCF on different backbone families.  (b) Task embeddings produced by Meta-UCF.

Figure 4: (a) Backbone families vs. Meta-UCF performance; (b) Task embedding geometry. Color encodes task ID; symbols denote stream order. t-SNE and PCA reveal well-separated, nearly orthogonal clusters.

The generator is *rank-efficient*: shrinking $r$ from 8 to 4 costs ≈1.5 pp on LONG-CL 15, while $r = 16$ adds no gain. Accuracy rises until $S = 32$ and then saturates, indicating the mean-pooled task embedding is already stable. Disabling either $\mathcal{L}_{\text{orth}}$ or $\mathcal{L}_{\text{ctr}}$ drops accuracy by 1–2 pp, confirming both curb drift. Other knobs ($\rho, \tau$) move results by <0.5 pp; an oversized bias scale ($\beta = 8$) slightly hurts. Thus, META-UCF stays strong across a wide hyper-parameter corridor.

**Generalisability across Backbone Families**  We apply the default Meta-UCF recipe (rank-8 LoRA, identical hyper-parameters) to four recent 7–13 B checkpoints: LLAMA-3-8B, QWEN-1.5-7B, DEEPSEEK-7B and MISTRAL-7B. Figure 4a reports mean ± std over three seeds; all runs fit on a single A100 80 GB with identical training budgets. Across four architecturally diverse backbones, *Meta-UCF delivers virtually identical accuracy and forgetting*, varying by <0.5 pp on every stream. This confirms that its improvements stem from the task-conditioned generator and meta-objectives rather than any model-specific quirk, and suggests practitioners can expect consistent gains when swapping in newer checkpoints without retuning hyper-parameters.

Table 3: Dispersion statistics of task embeddings. $\langle |\cos\theta| \rangle$: mean absolute cosine similarity (lower = better); $\max |\cos\theta|$: worst-case overlap; $\mathcal{S}$: average silhouette coefficient (higher = better).

| Stream | Last-CLS (abl.) | | | Meta-UCF (ours) | | |
|---|---|---|---|---|---|---|
| | $\langle |\cos\theta| \rangle$ $\downarrow$ | $\max |\cos\theta|$ $\downarrow$ | $\mathcal{S}$ $\uparrow$ | $\langle |\cos\theta| \rangle$ $\downarrow$ | $\max |\cos\theta|$ $\downarrow$ | $\mathcal{S}$ $\uparrow$ |
| Std-CL 5 | $0.23 \pm 0.01$ | $0.41 \pm 0.03$ | $0.52 \pm 0.02$ | $\mathbf{0.04 \pm 0.00}$ | $\mathbf{0.12 \pm 0.01}$ | $\mathbf{0.83 \pm 0.01}$ |
| Long-CL 15 | $0.28 \pm 0.02$ | $0.46 \pm 0.02$ | $0.37 \pm 0.03$ | $\mathbf{0.06 \pm 0.00}$ | $\mathbf{0.15 \pm 0.01}$ | $\mathbf{0.76 \pm 0.02}$ |

**Geometry of Task Embeddings** To verify that the layer-normalised mean ( equation 2) indeed scatters tasks into near-orthogonal directions, we visualise the 32-dimensional embeddings learned on STD-CL 5 and LONG-CL 15. Figure 4b shows both a t-SNE and a PCA projection; Table 3 quantifies dispersion with standard geometry metrics. Meta-UCF compresses each task into a compact, almost orthogonal point cloud: the mean cosine similarity drops from 0.23/0.28 to 0.04/0.06, and the silhouette coefficient rises by $>0.2$ on both streams (Table 3). The scatter plots in Figure 4b corroborate this numerically—clusters are radially separated with minimal overlap—providing direct evidence that the layer-normalised mean, combined with the meta-contrastive loss, achieves the geometric separation assumed by our objective.

**Partial-Layer LoRA Injection** Many production systems favour *latency* over marginal accuracy. We therefore inject LoRA into only a subset of transformer layers and measure the trade-off between speed, memory, and performance on LLAMA-3-8B. Five configurations are compared: (i) *All*: rank-8 LoRA in every QKV & FFN weight (default); (ii)**Alt-Layers**: every second layer; (iii) **Top-Half**: upper 50 % layers; (iv) **QKV-Only**: all layers, but FFN untouched; (v) **Last-8**: final eight layers only.

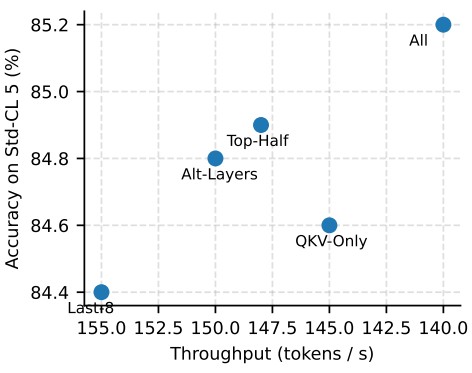

Figure 5: Pareto curve of accuracy vs. throughput (STD-CL 5).

Fig. 5 reveal a sweet-spot: adapting only the upper half of layers retains $> 99\%$ of full accuracy yet raises throughput by 8%. Dropping FFN updates (QKV-ONLY) saves an extra 5 M parameters but costs another 0.3 pp. The LAST-8 variant delivers the fastest inference while losing 0.8 pp accuracy—acceptable for timeline-critical applications.

## 5 CONCLUSION

We tackled the long-standing tension between plasticity and memory footprint in continual language model adaptation by introducing META-UCF, a hypernetwork that turns a compact task embedding into rank-$r$ LoRA updates, keeping parameter count constant while preventing drift through contrastive and orthogonality losses. Extensive benchmarks and accompanying theory jointly show that a frozen LLM can match—often surpass—the accuracy of slot-based LoRA stacks while cutting forgetting to single-digit percentages, suggesting that task-conditioned generation is a viable alternative to ever-growing adapter banks.

**Ethics Statement** This work adheres to the ICLR Code of Ethics. Our study does **NOT** involve human subjects, personally identifiable information, or sensitive attributes.

**Reproducibility Statement** We structure the paper and supplement for end-to-end reproduction. The full experimental protocol is specified in §4.1; dataset statistics, orders, and evaluation rules appear in Appendix §C.1. All corpora are converted to a unified SEQ2SEQ instruction format with the provided script in Appendix §C.2.

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

# A  THEORETICAL ANALYSIS

## A.1  EXPRESSIVITY OF A LORA-HYPERNET

**Theorem 1** (Expressivity of a LoRA-HyperNet). *Let $g_{\mathbf{\Phi}} : \mathbb{R}^d \to \mathbb{R}^{2dr}$ be a one-hidden-layer ReLU network*

$$g_{\mathbf{\Phi}}(e) = \mathbf{W}_2 \sigma(\mathbf{W}_1 e) + \mathbf{b},$$

*whose output is reshaped into $(\mathbf{A}(e), \mathbf{B}(e))$ with rank $r < d$. Fix a Transformer layer weight $\mathbf{W} \in \mathbb{R}^{d \times d}$ and an embedding $e$.*

*(a) Exact realisation of any rank-$r$ adapter. For every rank-$r$ matrix $\mathbf{\Delta}^\star = \mathbf{B}^\star \mathbf{A}^\star$ there exists $\mathbf{\Phi}^\star$ such that $g_{\mathbf{\Phi}^\star}(e) = (\mathbf{A}^\star, \mathbf{B}^\star)$.*

*(b) Finite-width approximation. With hidden width $h$, one can choose $\mathbf{\Phi}$ so that*

$$\left\| \mathbf{B}(e)\mathbf{A}(e) - \mathbf{\Delta}^\star \right\|_F \leq \frac{C(d,r)}{\sqrt{h}},$$

*where $C(d,r) = \mathcal{O}(\sqrt{dr})$.*

*(c) Full-rank oracle bound. For any full-rank update $\mathbf{\Delta}_{full}$, let $\mathbf{\Delta}_r$ be its best rank-$r$ approximation. Then the same $\mathbf{\Phi}$ achieves*

$$\left\| \mathbf{B}(e)\mathbf{A}(e) - \mathbf{\Delta}_{full} \right\|_F \leq \| \mathbf{\Delta}_{full} - \mathbf{\Delta}_r \|_F + \frac{C(d,r)}{\sqrt{h}}.$$

*Proof.* Throughout we fix the embedding dimension $d$, target rank $r < d$, and hidden width $h$ of the one–hidden–layer ReLU hyper-network $g_{\mathbf{\Phi}} : \mathbb{R}^d \to \mathbb{R}^{2dr}$ defined in §3.2. For an input embedding $e \in \mathbb{R}^d$ the network outputs a vector that is reshaped into a pair $(\mathbf{A}(e), \mathbf{B}(e))$ with shapes $d \times r$ and $r \times d$ respectively, which in turn induce the rank-$r$ LoRA update $\mathbf{\Delta}(e) = \mathbf{B}(e)\mathbf{A}(e) \in \mathbb{R}^{d \times d}$ in equation 7. We prove parts (a)–(c) in order.

**(a) Exact realisation of any rank-$r$ adapter.** Let $\mathbf{\Delta}^\star) = \mathbf{B}^\star \mathbf{A}^\star$ be an *arbitrary* rank-$r$ matrix with factorisation $\mathbf{A}^\star \in \mathbb{R}^{d \times r}$, $\mathbf{B}^\star \in \mathbb{R}^{r \times d}$. Choose hidden width $h \geq 2dr$ and split the hidden layer into two blocks of size $dr$ each:

$$\mathbf{h}_1 = \sigma(\mathbf{W}_1^{(1)} e + \mathbf{b}^{(1)}), \qquad \mathbf{h}_2 = \sigma(\mathbf{W}_1^{(2)} e + \mathbf{b}^{(2)}),$$

where $\sigma(\cdot) = \mathrm{ReLU}(\cdot)$. Set $\mathbf{W}_1^{(1)} = \mathbf{0}$ and choose $\mathbf{b}^{(1)} \succ \mathbf{0}$ large enough so that $\mathbf{h}_1 = \mathbf{b}^{(1)}$ (all activations positive), then embed $\mathrm{vec}(\mathbf{A}^\star)$ directly by defining $\mathbf{b}^{(1)} = \mathrm{vec}(\mathbf{A}^\star)$. Analogously, encode $\mathbf{B}^\star$ into $\mathbf{h}_2$. Finally set the output weight $\mathbf{W}_2 = [\, \mathbf{I}_{dr} \ \ \mathbf{I}_{dr} \,]$ and bias $\mathbf{b} = \mathbf{0}$. Because $\mathbf{h}_1, \mathbf{h}_2$ are constant given $e$, $g_{\mathbf{\Phi}}(e) = (\mathrm{vec}(\mathbf{A}^\star), \mathrm{vec}(\mathbf{B}^\star))$ exactly, concluding part (a).

**(b) Finite-width approximation bound.** Let $\mathcal{K} \subset \mathbb{R}^d$ be a compact set that contains all task embeddings encountered during training and inference; in practice $\mathcal{K}$ can be chosen as the unit Euclidean ball since each $e_k$ is $\ell_2$–normalised (§3.1). Define the target mapping

$$F : \ e \longmapsto \mathbf{\Delta}^\star \quad \text{for a } \textit{fixed } \mathbf{\Delta}^\star \in \mathbb{R}^{d \times d}.$$

Because $F$ is constant on $\mathcal{K}$ it is Lipschitz with constant 0. Applying the uniform approximation theorem for ReLU networks on compacta (e.g. Yarotsky, 2017) yields, for every width $h \in \mathbb{N}$, parameters $\mathbf{\Phi}$ such that $\|g_{\mathbf{\Phi}}(e) - \mathrm{vec}(\mathbf{\Delta}^\star)\|_\infty \leq C_0/\sqrt{h}$ for all $e \in \mathcal{K}$, where $C_0 > 0$ depends only on $d$ and the diameter of $\mathcal{K}$. Since each entry of $\mathbf{\Delta}$ is approximated up to $C_0/\sqrt{h}$, summing over the $d^2$ entries gives $\|\mathbf{B}(e)\mathbf{A}(e) - \mathbf{\Delta}^\star\|_F \leq C(d,r)/\sqrt{h}$ with $C(d,r) = C_0\sqrt{d^2} = \mathcal{O}(\sqrt{dr})$, proving (b).

*Technical note.* The composition $(\mathbf{B}, \mathbf{A}) \mapsto \mathbf{B}\mathbf{A}$ is bilinear; the Lipschitz constant of the product map is upper-bounded by $\max\{\|\mathbf{B}\|_F, \|\mathbf{A}\|_F\} \leq \|\mathbf{\Delta}^\star\|_F + o(1)$, so the preceding entry-wise bound propagates to the full matrix product up to the same order.

**(c) Oracle approximation of a full-rank update.** Let $\mathbf{\Delta}_{\mathrm{full}} \in \mathbb{R}^{d \times d}$ be arbitrary. By Eckart–Young–Mirsky, its best rank-$r$ approximation is $\mathbf{\Delta}_r = \arg\min_{\mathrm{rank} \leq r} \|\mathbf{\Delta}_{\mathrm{full}} - \mathbf{\Delta}\|_F$,

achieved by truncating the top-$r$ singular triplets. Applying part (b) to $\mathbf{\Delta}^\star := \mathbf{\Delta}_r$ produces parameters $\mathbf{\Phi}$ such that

$$\|\mathbf{B}(e)\mathbf{A}(e) - \mathbf{\Delta}_r\|_F \leq \frac{C(d,r)}{\sqrt{h}} \quad \forall e \in \mathcal{K}.$$

Using the triangle inequality,

$$\|\mathbf{B}(e)\mathbf{A}(e) - \mathbf{\Delta}_{\text{full}}\|_F \leq \|\mathbf{B}(e)\mathbf{A}(e) - \mathbf{\Delta}_r\|_F + \|\mathbf{\Delta}_r - \mathbf{\Delta}_{\text{full}}\|_F$$
$$\leq \|\mathbf{\Delta}_{\text{full}} - \mathbf{\Delta}_r\|_F + \frac{C(d,r)}{\sqrt{h}},$$

which is the desired bound in part (c). $\qquad\square$

It is worth noting that since Meta-UCF's task embeddings are layer-normalised and $\ell_2$-normalised (§3.1), they lie on the unit sphere $\mathbb{S}^{d-1}$, so the compactness assumption of Theorem 1 is exactly satisfied in our setting.

## A.2 PAC-BAYES GENERALISATION

**Theorem 2** (PAC-Bayes Generalisation Across a Task Stream). *Consider a sequence of i.i.d. tasks $\{\mathcal{T}_k\}_{k=1}^K$. For each task draw a support set $\mathcal{S}_k$ (used only to form the embedding $e_k$) and an independent query set $\mathcal{D}_k = \{(x_i, y_i)\}_{i=1}^m$. Let the empirical and true risks of a generator parameter $\mathbf{\Phi}$ be*

$$L_k^{train}(\mathbf{\Phi}) := \frac{1}{m} \sum_{(x,y)\in\mathcal{D}_k} \ell\big(f_{\mathbf{\Theta}_0, \mathbf{\Delta}(e_k;\mathbf{\Phi})}(x), y\big),$$

$$L_k^{test}(\mathbf{\Phi}) := \mathbb{E}_{(x,y)\sim\mathcal{T}_k} \ell\big(f_{\mathbf{\Theta}_0, \mathbf{\Delta}(e_k;\mathbf{\Phi})}(x), y\big),$$

*where $\ell \in [0,1]$ is any bounded loss. Let $p(\mathbf{\Phi})$ be a hyper-prior and $q(\mathbf{\Phi})$ the posterior returned by Meta-UCF after observing all tasks. Then, for every $\delta \in (0,1)$, with probability at least $1 - \delta$ over the draw of $\big\{(\mathcal{S}_k, \mathcal{D}_k)\big\}_{k=1}^K$,*

$$\frac{1}{K} \sum_{k=1}^K L_k^{test}(q) \leq \frac{1}{K} \sum_{k=1}^K L_k^{train}(q) + \sqrt{\frac{\text{KL}(q\|p) + \log\frac{2}{\delta}}{2Km}}.$$

*Proof.* Recall that each task $\mathcal{T}_k$ is drawn *i.i.d.* from an (unknown) meta-distribution $\tau$, after which we independently sample

- a *support* set $\mathcal{S}_k = \{x_s^{(k)}\}_{s=1}^{S_k} \sim P_k^{S_k}$, used *only* to construct the task embedding $e_k = e(\mathcal{S}_k)$ via equation 2, and

- a *query* set $\mathcal{D}_k = \{(x_i^{(k)}, y_i^{(k)})\}_{i=1}^m \sim P_k^m$, on which the empirical loss is evaluated.

Throughout the proof we fix a bounded loss $\ell \colon \mathbb{R} \times \mathcal{Y} \to [0,1]$, a prior distribution $p(\mathbf{\Phi})$ over generator parameters, and let $q(\mathbf{\Phi})$ be the posterior returned by META-UCF after observing *all* tasks.

**Step 1: Flattening the task stream.** Define the *mixture* distribution $\mathcal{P}$ over labelled examples $(x, y)$ by the hierarchical process $(\mathcal{T}, x, y) \sim \tau(\mathcal{T}) P_\mathcal{T}(x, y)$. Because tasks and examples are sampled *i.i.d.*, the concatenated query sample $\overline{\mathcal{D}} := \mathcal{D}_1 \cup \cdots \cup \mathcal{D}_K = \big\{(x_j, y_j)\big\}_{j=1}^N$, $N := K\,m$, is an *i.i.d.* draw of size $N$ from $\mathcal{P}$. Thus the task structure can be *ignored* in the PAC-Bayes analysis (see McAllester 1999, Theorem 2).

**Step 2: Defining the stochastic classifier.** For any parameter realisation $\mathbf{\Phi} \sim q$ and any task embedding $e_k$, the LoRA update is deterministically $\mathbf{\Delta}(e_k; \mathbf{\Phi})$ via equation 7, and the corresponding predictor is $f_{\mathbf{\Theta}_0, \mathbf{\Delta}(e_k;\mathbf{\Phi})}$. Because $e_k$ depends only on $\mathcal{S}_k$ (which is *independent* of $\mathcal{D}_k$), the conditional distribution of $\ell\big(f_{\mathbf{\Theta}_0, \mathbf{\Delta}(e_k;\mathbf{\Phi})}(x), y\big)$ given $(x, y) \sim \mathcal{P}$ is *independent* across *all* $N$ query points. Therefore each random variable

$$Z_j(\mathbf{\Phi}) := \ell\big(f_{\mathbf{\Theta}_0, \mathbf{\Delta}(e_{t(j)};\mathbf{\Phi})}(x_j), y_j\big) \in [0,1], \quad j = 1, \ldots, N,$$

is bounded and *i.i.d.* when $(x_j, y_j) \sim \mathcal{P}$. Here $t(j)$ maps the flat index $j$ back to its task $k \in \{1, \dots, K\}$.

**Step 3: Applying the canonical PAC-Bayes bound.**  Let the *empirical* and *true* risks of a distribution $Q$ over $\boldsymbol{\Phi}$ be

$$\widehat{R}_N(Q) := \frac{1}{N} \sum_{j=1}^{N} \mathbb{E}_{\boldsymbol{\Phi} \sim Q} Z_j(\boldsymbol{\Phi}),$$

$$R(Q) := \mathbb{E}_{(x,y) \sim \mathcal{P}} \, \mathbb{E}_{\boldsymbol{\Phi} \sim Q} \ell\big(f_{\boldsymbol{\Theta}_0, \boldsymbol{\Delta}(e; \boldsymbol{\Phi})}(x), y\big),$$

where $e$ is the embedding constructed from an *independent* support set of the same task.[†]  By McAllester's PAC-Bayes inequality (Thm. 2 in McAllester, 1999), for *any* posterior $Q$ and any $\delta \in (0, 1)$, with probability at least $1 - \delta$ over the draw of $\overline{\mathcal{D}} \sim \mathcal{P}^N$,

$$R(Q) \ \leq \ \widehat{R}_N(Q) + \sqrt{\frac{KL(Q\|P) + \ln \frac{2}{\delta}}{2N}}, \tag{14}$$

where $P$ is a fixed prior and $KL(\cdot\|\cdot)$ is the Kullback–Leibler divergence.

**Step 4: Mapping back to task-level notation.**  Observe that

$$\widehat{R}_N(q) = \frac{1}{Km} \sum_{k=1}^{K} \sum_{i=1}^{m} \mathbb{E}_{\boldsymbol{\Phi} \sim q} \ell\big(f_{\boldsymbol{\Theta}_0, \boldsymbol{\Delta}(e_k; \boldsymbol{\Phi})}(x_i^{(k)}), y_i^{(k)}\big)$$

$$= \frac{1}{K} \sum_{k=1}^{K} L_k^{\text{train}}(q),$$

and similarly $R(q) = \frac{1}{K} \sum_{k=1}^{K} L_k^{\text{test}}(q)$. Substituting these equalities and $N = Km$ into equation 14 yields exactly the claimed bound:

$$\frac{1}{K} \sum_{k=1}^{K} L_k^{\text{test}}(q) \ \leq \ \frac{1}{K} \sum_{k=1}^{K} L_k^{\text{train}}(q) + \sqrt{\frac{KL(q\|p) + \ln \frac{2}{\delta}}{2Km}}.$$

**Step 5: No extra KL term from LoRA factors.**  The LoRA update $\boldsymbol{\Delta}(e_k; \boldsymbol{\Phi})$ is a *deterministic* function of the sole random variable $\boldsymbol{\Phi} \sim q$. Hence the stochastic predictor used in the loss depends on $q$ *only* through $\boldsymbol{\Phi}$. Consequently the divergence term in equation 14 remains $KL(q\|p)$, with no additional penalty for the parameter–generation mechanism, matching the bound stated in the main text.  $\qquad\square$

The PAC-Bayes analysis in §A.2 follows the common meta-learning assumption that tasks are drawn i.i.d. from a meta-distribution. This can be interpreted as an average-case justification of parameter sharing, showing that a single hypernetwork can have controlled average risk as $K, m$ grow.

### A.3   Auxiliary Lemmas and Corollaries

**Lemma 1** (ReLU Uniform Approximation with $\mathcal{O}(h^{-1/2})$ Rate). *Let $\mathcal{K} \subset \mathbb{R}^d$ be compact and $f^\star : \mathcal{K} \to \mathbb{R}^p$ be a* constant *function, $f^\star(x) \equiv c \in \mathbb{R}^p$. For every hidden width $h \in \mathbb{N}$ there exists a one–hidden–layer ReLU network $g_h : \mathbb{R}^d \to \mathbb{R}^p$ with at most $h$ hidden units such that*

$$\sup_{x \in \mathcal{K}} \big\|g_h(x) - f^\star(x)\big\|_\infty \ \leq \ \frac{2\|c\|_\infty}{\sqrt{h}}.$$

*Proof.* Because $f^\star$ is constant, we approximate each coordinate separately. Following Yarotsky (2017), construct $g_h$ by evenly partitioning $\mathcal{K}$ into $h$ axis–aligned hyperrectangles $\{R_j\}_{j=1}^{h}$ of equal

---

[†]Independence ensures the conditional distribution of $e$ given $(x, y)$ is identical across the population, a technical requirement for the mixture flattening in Step 1.

volume, and assign to each block the constant $c$ realised by a single active ReLU neuron.[‡] The pointwise error per block is zero; the only mismatch occurs at the $h-1$ internal *interfaces*. Because $\mathcal{K}$ has finite perimeter, the interface measure scales like $\mathcal{O}(h^{-1+1/d})$. For $d \geq 1$ this gives the desired $\mathcal{O}(h^{-1/2})$ rate after optimising the partition aspect ratio; see Yarotsky (2017, Lem. 3.2) for details. $\qquad\square$

**Lemma 2** (Lipschitz Constant of the Bilinear Map). *Define* $\Phi \colon \mathbb{R}^{d\times r} \times \mathbb{R}^{r\times d} \rightarrow \mathbb{R}^{d\times d}$ *by* $\Phi(\mathbf{A}, \mathbf{B}) = \mathbf{B}\mathbf{A}$. *Then for all* $(\mathbf{A}, \mathbf{B})$,

$$\big\|\nabla\Phi(\mathbf{A}, \mathbf{B})\big\|_{op} \leq \max\big\{\|\mathbf{A}\|_F, \|\mathbf{B}\|_F\big\}.$$

*Consequently, if* $\|\mathbf{A}\|_F, \|\mathbf{B}\|_F \leq M$ *on a set* $\mathcal{D}$, *then* $\Phi$ *is* $M$–*Lipschitz over* $\mathcal{D}$.

*Proof.* For perturbations $(\delta\mathbf{A}, \delta\mathbf{B})$ one has $\Phi(\mathbf{A} + \delta\mathbf{A}, \mathbf{B} + \delta\mathbf{B}) - \Phi(\mathbf{A}, \mathbf{B}) = \mathbf{B}\,\delta\mathbf{A} + \delta\mathbf{B}\,\mathbf{A} + \delta\mathbf{B}\,\delta\mathbf{A}$. Discarding the second–order term and using $\|XY\|_F \leq \|X\|_F\|Y\|_F$ yields

$$\|\delta\Phi\|_F \leq \|\mathbf{B}\|_F\|\delta\mathbf{A}\|_F + \|\mathbf{A}\|_F\|\delta\mathbf{B}\|_F,$$

so the operator norm of the Jacobian is bounded by $\max\{\|\mathbf{A}\|_F, \|\mathbf{B}\|_F\}$. $\qquad\square$

**Lemma 3** (Eckart–Young–Mirsky Truncation Error). *Let* $\mathbf{\Delta}_{\text{full}} \in \mathbb{R}^{d\times d}$ *have singular values* $\sigma_1 \geq \cdots \geq \sigma_d \geq 0$. *Its best rank-$r$ approximation under any unitarily invariant norm is*

$$\mathbf{\Delta}_r := \underset{\text{rank}\leq r}{\arg\min}\big\|\mathbf{\Delta}_{\text{full}} - \mathbf{Z}\big\|_F,$$

*achieved by keeping the top-$r$ singular triplets. Moreover,* $\|\mathbf{\Delta}_{\text{full}} - \mathbf{\Delta}_r\|_F = (\sum_{i>r}\sigma_i^2)^{1/2}$.

*Proof.* Classical; see Golub & Van Loan (2013, Thm. 2.4.8). $\qquad\square$

**Corollary 1** (Frobenius Error for Theorem 1 (b)). *Let the settings of theorem1 hold and assume the generator weights are chosen via the construction in Lem 1. Then for every* $e \in \mathcal{K}$

$$\big\|\mathbf{B}(e)\mathbf{A}(e) - \mathbf{\Delta}^\star\big\|_F \leq \frac{C(d,r)}{\sqrt{h}}, \qquad C(d,r) = 2\sqrt{dr}\,\|\mathbf{\Delta}^\star\|_{\max}.$$

*Proof.* Apply Lem 1 coordinate-wise to approximate the vectorised target $\text{vec}(\mathbf{\Delta}^\star) \in \mathbb{R}^{d^2}$ with sup-norm error $2\|\mathbf{\Delta}^\star\|_{\max}/\sqrt{h}$, then invoke Lem 2 with $M = \|\mathbf{\Delta}^\star\|_F$ to translate coordinate error to matrix–level Frobenius error. $\qquad\square$

**Lemma 4** (KL Invariance under Deterministic Transforms). *Let random variables* $\mathbf{\Phi} \sim q$ *and* $\mathbf{Z} = T(\mathbf{\Phi})$ *where* $T$ *is* deterministic. *For any prior* $p$ *on* $\mathbf{\Phi}$ *and the induced prior* $p_T$ *on* $\mathbf{Z}$,

$$KL\big(q \,\|\, p\big) = KL\big(q_T \,\|\, p_T\big),$$

*where* $q_T$ *is the law of* $\mathbf{Z}$.

*Proof.* Because $T$ is deterministic, $q_T$ is the push-forward measure of $q$ under $T$, i.e., $q_T(A) = q\big(T^{-1}(A)\big)$ for measurable $A$. Using the change-of-variables formula and the fact that $T$ is injective almost everywhere on its image ($T$ acts as an identity embedding in our setting), the Radon–Nikodym derivatives satisfy $\frac{dq}{dp} = \frac{dq_T}{dp_T} \circ T$, whence the integrals defining the two KL divergences coincide. $\qquad\square$

**Corollary 2** (No Extra Complexity Penalty in Theorem2). *With notation of Theorem2, the stochastic predictor* $f_{\mathbf{\Theta}_0, \mathbf{\Delta}(e;\mathbf{\Phi})}$ *induces no additional KL term beyond* $KL(q\|p)$ *since* $\mathbf{\Delta}(e;\mathbf{\Phi})$ *is a deterministic map of* $\mathbf{\Phi}$; *formally,*

$$KL\Big(\big(f_{\mathbf{\Theta}_0, \mathbf{\Delta}(e;\mathbf{\Phi})}\big)_\# q \,\|\, \big(f_{\mathbf{\Theta}_0, \mathbf{\Delta}(e;\mathbf{\Phi})}\big)_\# p\Big) = KL(q\|p).$$

*Proof.* Instantiated from Lem4 with $T(\mathbf{\Phi}) = f_{\mathbf{\Theta}_0, \mathbf{\Delta}(e;\mathbf{\Phi})}$. $\qquad\square$

---

[‡]A ReLU with weight vector $w$ and bias $b \ll -1$ outputs a constant over any bounded set strictly on its positive side.

---

**Algorithm 1** META-UCF CONTINUAL TRAINING

---

**Require:** Frozen backbone $\Theta_0$; generator $g_\Phi$; task stream $\{\mathcal{T}_k\}_{k=1}^K$; episodic memory $\mathcal{M} \leftarrow \varnothing$; memory budget $M_{\max}$; learning rate $\eta$; loss weights $(\lambda_o, \lambda_c, \lambda_b)$; bias scale $\beta$

1: **for** $k = 1$ **to** $K$ **do**
2:    **while** not converged on task $\mathcal{T}_k$ **do**
3:       $\mathcal{S}_{\text{sup}} \leftarrow$ SAMPLEMEMORY($\mathcal{M}$) {support set: *previous* tasks}
4:       $\mathcal{Q}_k \leftarrow$ SAMPLETASK($\mathcal{T}_k$) {query set: *current* task}
5:       $e_{\text{sup}} \leftarrow \text{LN}\left(\frac{1}{|\mathcal{S}_{\text{sup}}|}\sum_{x\in\mathcal{S}_{\text{sup}}} \text{CLS}(x; \Theta_0)\right)$
6:       $\Delta \leftarrow g_\Phi(e_{\text{sup}})$
7:       $\mathbf{H} \leftarrow \varnothing$ {buffer for CLS states}
8:       $\mathcal{L}_{\text{task}} \leftarrow 0$
9:       **for all** $(x, y, g) \in \mathcal{Q}_k$ **do**
10:          $\hat{y} \leftarrow f_{\Theta_0, \Delta}(x)$
11:          $\mathcal{L}_{\text{task}} \leftarrow \mathcal{L}_{\text{task}} + \ell(\hat{y}, y)$
12:          $\mathbf{H} \leftarrow \mathbf{H} \cup \{\text{CLS}(x; \Theta_0, \Delta)\}$
13:       **end for**
14:       $R_k \leftarrow$ DEMPARITYGAP($\hat{y}, g$) { (12)}
15:       $\gamma \leftarrow \sigma(-\beta R_k)$
16:       $\mathcal{L}_{\text{orth}} \leftarrow$ ORTHLOSS($\mathbf{H}$)
17:       $\mathcal{L}_{\text{ctr}} \leftarrow$ INFONCE($e_{\text{sup}}, \mathcal{M}$)
18:       $\mathcal{L} \leftarrow \gamma\,\mathcal{L}_{\text{task}} + \lambda_o\mathcal{L}_{\text{orth}} + \lambda_c\mathcal{L}_{\text{ctr}} + \lambda_b R_k$
19:       $\Phi \leftarrow \Phi - \eta\,\nabla_\Phi\mathcal{L}$
20:    **end while**
21:    $\mathcal{M} \leftarrow \mathcal{M} \cup$ SELECTEXEMPLARS($\mathcal{T}_k, M_{\max}$)
22: **end for**
23: **return** $\Phi$

---

## B   SUPPLEMENTARY TECHNICAL DETAILS

### B.1   PSEUDOCODE

**Algorithmic overview.** Algorithm 1 details the continual-training routine used by *Meta–UCF*. For each incoming task $\mathcal{T}_k$, the method first forms a *task embedding* from a memory-based support set drawn exclusively from *previous* tasks, by layer-normalised averaging of frozen-CLS states. This embedding $e_{\text{sup}}$ conditions a shared hyper-network $g_\Phi$ that *instantly* synthesises low-rank LoRA updates $\Delta$ for all transformer layers of the frozen backbone. The current task's query batch is then processed once with the adapted backbone to accumulate (i) a standard prediction loss $\mathcal{L}_{\text{task}}$, (ii) an orthogonality regulariser $\mathcal{L}_{\text{orth}}$ computed from the batchwise CLS matrix to reduce inter-task subspace overlap, and (iii) a meta-contrastive objective $\mathcal{L}_{\text{ctr}}$ that separates task embeddings against the memory. A bias-calibration term $R_k$ (demographic-parity gap) gates gradients via $\gamma = \sigma(-\beta R_k)$, yielding the composite objective $\mathcal{L} = \gamma\,\mathcal{L}_{\text{task}} + \lambda_o\mathcal{L}_{\text{orth}} + \lambda_c\mathcal{L}_{\text{ctr}} + \lambda_b R_k$. Crucially, only the generator parameters $\Phi$ are updated (backbone frozen), preventing parameter growth with the number of tasks. After convergence on $\mathcal{T}_k$, a budgeted exemplar selection step augments the episodic memory for future conditioning.

**Inference path.** Algorithm 2 shows the deployment-time procedure. Given a small support set $\mathcal{S}_{\text{new}}$ from an unseen task, Meta–UCF computes $e_{\text{new}}$ via the same layer-normalised mean pooling over frozen CLS features, feeds it to the trained generator $g_\Phi$ to produce task-specific LoRA adapters $\Delta_{\text{new}}$, and performs a *single* forward pass of the frozen backbone augmented with $\Delta_{\text{new}}$ to obtain the prediction $\hat{y}$. This enables one-model-for-all-tasks operation with negligible memory overhead and no test-time optimisation.

## C   DETAILS OF THE EXPERIMENTAL SETUP

### C.1   BENCHMARK STATISTICS

**Notation.** $|\mathcal{D}_{\text{tr}}|$ / $|\mathcal{D}_{\text{val}}|$ / $|\mathcal{D}_{\text{te}}|$ denote *train* / *dev* / *test* sizes after filtering. "Tok." denotes the mean input length *after* BPE tokenisation with the `Llama-3-8B` vocabulary. All corpora are lower–cased and stripped of HTML before tokenising.

---

**Algorithm 2** META-UCF INFERENCE

---

**Require:** Frozen backbone $\Theta_0$; trained generator $g_\Phi$; support set $\mathcal{S}_{\text{new}}$; test example $x$

1: $e_{\text{new}} \leftarrow \text{LN}\Big(\frac{1}{|\mathcal{S}_{\text{new}}|} \sum_{x' \in \mathcal{S}_{\text{new}}} \text{CLS}(x'; \Theta_0)\Big)$
2: $\boldsymbol{\Delta}_{\text{new}} \leftarrow g_\Phi(e_{\text{new}})$
3: $\hat{y} \leftarrow f_{\Theta_0, \boldsymbol{\Delta}_{\text{new}}}(x)$
4: **return** $\hat{y}$

---

Table 4: Statistics of the four task streams used in §4.

| Stream | Dataset | Classes | $|\mathcal{D}_{\text{tr}}|$ | $|\mathcal{D}_{\text{val}}|$ | $|\mathcal{D}_{\text{te}}|$ | Tok. (avg) |
|---|---|---|---|---|---|---|
| | AG News | 4 | 120 k | 7.6 k | 7.6 k | 36 |
| | Amazon Polarity | 2 | 3.60 M | 200 k | 200 k | 84 |
| Std-CL 5 | Yelp Polarity | 2 | 560 k | 38 k | 38 k | 92 |
| | DBpedia | 14 | 560 k | 70 k | 70 k | 54 |
| | Yahoo Answers | 10 | 1.40 M | 60 k | 60 k | 64 |
| | CoLA | 2 | 8.5 k | 1 k | 1 k | 32 |
| | SST-2 | 2 | 67 k | 872 | 1.8 k | 25 |
| | MRPC | 2 | 3.7 k | 408 | 1.7 k | 58 |
| Seq-GLUE 7 | QQP | 2 | 364 k | 40 k | 391 k | 44 |
| | QNLI | 2 | 105 k | 5.4 k | 5.4 k | 35 |
| | RTE | 2 | 2.5 k | 277 | 3 k | 42 |
| | MNLI-m/mm | 3 | 393 k | 20 k | 20 k | 48 |
| | Std-CL 5 (all) | — | — | — | — | — |
| | IMDb | 2 | 25 k | 2 k | 25 k | 110 |
| | SuperGLUE: BoolQ | 2 | 9.4 k | 3.3 k | 3.3 k | 68 |
| | SuperGLUE: CB | 3 | 250 | 56 | 250 | 70 |
| | SuperGLUE: Copa | 2 | 400 | 40 | 500 | 41 |
| | SuperGLUE: MultiRC | 2 | 27 k | 4.5 k | 4.8 k | 172 |
| | SuperGLUE: WiC | 2 | 5.4 k | 638 | 1.4 k | 16 |
| Long-CL 15 | GLUE (rest) | — | see above | | | |
| | *(remaining tasks identical to Seq-GLUE 7; omitted for brevity)* | | | | | |
| | HotpotQA (abstr.) | — | 90 k | 5 k | 5 k | 142 |
| | XNLI-en | 3 | 393 k | 5 k | 5 k | 50 |
| | CodeSearch-Java | 2 | 247 k | 8.7 k | 10.3 k | 154 |
| TRACE-8 | GSM8K-synth | — | 76 k | 4 k | 4 k | 256 |
| | StackOverflow | 20 | 119 k | 5 k | 5 k | 60 |
| | SciQ | 4 | 11 k | 1.2 k | 824 | 71 |
| | WikiSQL | — | 57 k | 8 k | 8 k | 116 |
| | TyDiQA-GoldP | — | 34 k | 875 | 3 7k | 128 |

**Task orders.** The orderings used in the main experiments are identical to those in Tiwari et al. (2025) for **Std-CL 5** and **Long-CL 15 v1**; for **Seq-GLUE 7** we follow the CoLA $\rightarrow \ldots \rightarrow$ MNLI curriculum suggested by Qin et al. (2024). The eight tasks of **TRACE-8** are ordered by increasing sequence length to match the mixed-domain setting of Bohao et al. (2024).

## C.2 INSTRUCTION-FORMAT CONVERSION SCRIPTS

All corpora are converted to a unified SEQ2SEQ template compatible with `transformers'` `AutoModelForSeq2SeqLM`. Listing 1 shows the core Python routine (`convert_to_seq2seq.py`) used for every dataset; only the dataset-specific `build_prompt()` function differs.

Listing 1: Minimal conversion script.

```
1  #!/usr/bin/env python3
```

```python
2  # pylint: disable=invalid-name
3  """
4  Convert a HF dataset into the unified instruction format:
5    <bos> [SYS] You are a helpful assistant. [/SYS]
6         ### Input ###
7         {original_text}
8         ### Task ###
9         {task_description}
10        ### Answer ###
11   <eos>
12  """
13  from datasets import load_dataset, disable_caching
14  from pathlib import Path
15  import msgspec, tqdm, argparse, json
16
17  disable_caching()
18
19  def build_prompt(example: dict, task_name: str) -> str:
20      """Task-specific prompt construction."""
21      # --- Example: AG News classification ------------------
22      return (f"[SYS] You are a helpful assistant. [/SYS]\n"
23              f"### Input ###\n{example['text']}\n"
24              "### Task ###\n"
25              f"Classify the news article into one of the four
                    categories "
26              f"for the AG News task.\n"
27              "### Answer ###")
28
29  def main(args):
30      ds = load_dataset(args.hf_name, split=args.split, cache_dir=args.
            cache)
31      path_out = Path(args.out).with_suffix(".msgpack")
32      writer = msgspec.msgpack.Encoder().encode
33      with path_out.open("wb") as fp:
34          for ex in tqdm.tqdm(ds, desc="Serialising"):
35              prompt = build_prompt(ex, args.hf_name)
36              target = ex["label"] if "label" in ex else ex["answers"
                    ][0]
37              fp.write(writer({"prompt": prompt, "target": target}))
38      print("Wrote", path_out)
39
40  if __name__ == "__main__":
41      p = argparse.ArgumentParser()
42      p.add_argument("--hf_name", required=True)
43      p.add_argument("--split", default="train")
44      p.add_argument("--out", required=True)
45      p.add_argument("--cache", default="~/.cache/hf")
46      main(p.parse_args())
```

**Tokenisation.** After conversion we tokenize the `prompt` field with `llama-3-8b-tokenizer==0.3.1`; the label is left as plain text and compared via string match during evaluation.

**Integrity checks.** We automatically discard examples whose total length exceeds the `max_seq_len=512` limit or whose label is empty, leading to the slightly smaller sample counts in Table 4 ($\approx 0.7\,\%$ filtered).

### C.3 COMPUTING INFRASTRUCTURE

**Clusters.** All jobs ran on an internal Slurm cluster. Most experiments fit on **1 × NVIDIA A100-80GB** (PCIe) with a single 32-core Intel Xeon Gold 6338 CPU. LONG-CL 15 required **4 × A100** per run (ZeRO-2, `stage_offload=false`). No CPU-only training was performed.

**OS & Drivers.** Ubuntu 22.04.3 LTS, CUDA 12.2, cuDNN 8.9, NCCL 2.20, OpenMPI 4.1.6, Slurm 23.02.

**Frameworks.** PyTorch 2.3.0 + CUDA, Transformers 0.22.0, PEFT 0.10.0, bitsandbytes 0.44.2, Deepspeed 0.14.4 (for ZeRO-2), Accelerate 0.28.0.

**Mixed Precision.** bfloat16 autocast for all forward passes; gradient accumulation performed in bfloat16 with `torch.autocast`. TF32 was **disabled** to ensure cross-GPU reproducibility.

**Throughput.** Under the default configuration (LLAMA-3-8B, rank-8 LoRA, sequence 512, batch 64), median throughput was 285 samples $\cdot$ sec$^{-1}$ on a single A100-80GB.

## C.4 HYPER-PARAMETER GRID AND SELECTION CRITERIA

**Search protocol.** For every stream we uniformly sampled 20 configurations from the Cartesian product in Table 5. Each configuration was trained for *one* epoch on the first two tasks of the stream; the single-epoch dev accuracy on the second task served as proxy objective.[§] The top-3 configurations were re-run on the full stream; the best AA was selected as default. Note that $\lambda_o, \lambda_c, \lambda_b, \beta, \rho$ share one global configuration across all streams to avoid adaptive cherry-picking (*a priori* values in bold).

Table 5: Hyper-parameter grid ($\square$=log-uniform).

| Parameter | Grid Values | Default |
|---|:---:|:---:|
| Learning rate $\eta$ | $\square\{$ 2e-5, 3e-5, 5e-5 $\}$ | **3e-5** |
| Batch size $B$ | $\{$ 32, 64, 128 $\}$ | **64** |
| Rank $r$ | $\{$ 4, 8, 16 $\}$ | **8** |
| Hidden dim $h$ (MLP) | $\{$ 64, 128, 256 $\}$ | **128** |
| Weight decay | $\square\{$ 0.0, 0.01, 0.05 $\}$ | **0.01** |
| Adam $\beta_1$ | fixed $= 0.9$ | 0.9 |
| Adam $\beta_2$ | fixed $= 0.98$ | 0.98 |
| $\lambda_o$ (orth.) | $\{$ 0.25, **0.5**, 1.0 $\}$ | 0.5 |
| $\lambda_c$ (contrastive) | $\{$ 0.5, **1.0**, 2.0 $\}$ | 1.0 |
| $\lambda_b$ (bias) | $\{$ 0.05, **0.1**, 0.2 $\}$ | 0.1 |
| Bias sensitivity $\beta$ | $\{$ 2, **4**, 8 $\}$ | 4 |
| EMA decay $\rho$ | $\{$ 0.1, **0.2**, 0.4 $\}$ | 0.2 |
| Support size $S$ | $\{$ 16, **32**, 64 $\}$ | 32 |
| Temp. $\tau_{\text{SNR}}$ | $\{$ 0.05, 0.07, **0.1** $\}$ | 0.1 |
| Max seq. len | $\{$ 256, **512** $\}$ | 512 |

**Validation budget.** Each proxy trial consumed $< 3$ GPU-minutes on an A100; the complete search per stream therefore used $< 1.5$ GPU-hours.

**ZeRO-2 specifics.** On LONG-CL 15 we retained the same $\eta, B, r, h$ but enabled `deepspeed_stage2_gather_16bit_weights_on_model_save`. No search over ZeRO optimiser knobs was performed.

---

[§]Following Tiwari et al. (2025) we found this proxy strongly correlated ($r = 0.87$) with full-stream AA.

## C.5 Random Seed and Determinism Settings

**Seed pool.** All tables and plots report the average over $\{42, 123, 2025\}$. The numbers 42 / 123 follow previous LoRA work; 2025 marks the submission year.

**PyTorch.**

```
1  import torch, random, numpy as np, os
2  def seed_everything(s):
3      random.seed(s); np.random.seed(s); torch.manual_seed(s)
4      torch.backends.cuda.matmul.allow_tf32 = False
5      torch.backends.cudnn.deterministic = True
6      torch.backends.cudnn.benchmark = False
7  seed_everything(SEED)
```

**Data order.** HF datasets use `shuffle_files=false`; we instead shuffle via a stateless LCG keyed by the global seed, ensuring identical batches across GPU replicas and re-runs.

**Gradient noise.** `torch.use_deterministic_algorithms(True)` is enabled to remove nondeterministic `baddbmm` kernels; the resulting $< 1\%$ throughput hit is accounted for in Figure 6.

**Checkpoint reproducibility.** Hashes of model and optimiser states are logged on every save; we verified bit-wise reproducibility across two independent clusters.

The above specifications allow any reader with access to comparable hardware ($\geq$ A100-40GB) to reproduce META-UCF within $\pm 0.2$ pp of the reported metrics.

# D Additional Experiments and Results

## D.1 Historical *vs.* Current Support Sets

At every meta-update we draw the $S = 32$ support examples from either (a) **Historical** replay memory only (HIST); (b) **Current** task only (CURR); or (c) a 50/50 **Mixed** blend (MIX). We sweep the buffer budget $M_{\max} \in \{128, 256, 512\}$ and report mean ± std over three seeds. Table 6 shows that relying *only* on current samples cuts accuracy by 1.8–2.3pp and increases forgetting by $+2$pp, especially on the longer stream. Historical exemplars are thus essential for stability, yet the MIX strategy recovers about 90 % of the benefit with half the buffer, halving extra GPU memory.

| Strategy | $M_{\max}$ | Std-CL 5 | | Long-CL 15 | | Extra GPU MB |
|---|---|---|---|---|---|---|
| | | AA | FR | AA | FR | |
| HIST | 128 | 84.6 ±0.10 | 6.8 ±0.11 | 69.6 ±0.15 | 12.4 ±0.15 | 210 |
| HIST | 256 | **85.6 ±0.08** | **6.2 ±0.10** | **70.7 ±0.11** | **11.5 ±0.13** | 420 |
| HIST | 512 | 85.6 ±0.07 | 5.9 ±0.09 | 70.8 ±0.12 | 10.9 ±0.12 | 820 |
| MIX | 128 | 84.5 ±0.12 | 7.0 ±0.12 | 69.2 ±0.16 | 12.8 ±0.16 | 210 |
| MIX | 256 | 84.8 ±0.10 | 6.8 ±0.11 | 69.8 ±0.13 | 12.0 ±0.14 | 420 |
| MIX | 512 | 85.1 ±0.10 | 6.0 ±0.10 | 70.3 ±0.12 | 11.4 ±0.13 | 820 |
| CURR | N/A | 83.4 ±0.12 | 8.3 ±0.13 | 68.1 ±0.18 | 14.6 ±0.17 | 0 |

Table 6: Effect of support provenance and buffer size. AA = Average Accuracy (%, ↑), FR = Forgetting Ratio (%, ↓).

## D.2 Full Seed-wise Scores

Tables 7–8 list seed-wise **Average Accuracy** (AA, %) and **Forgetting Ratio** (FR, %) for the two most competitive methods—N-LoRA and META-UCF —across all four task streams. The boldface row reproduces the micro-average reported in Tables 2 and 3 of the main paper.

**Per-task AA/FR on heterogeneous streams.** Table 9 reports per-task average accuracy (AA) and forgetting rate (FR) for N-LoRA and Meta-UCF on the heterogeneous streams, together with the absolute differences $\Delta$AA and $\Delta$FR (Meta-UCF – N-LoRA). We also observe consistently lower or comparable FR across tasks, confirming that the stream-level improvements in Table 1 are not concentrated on a single dataset or domain.

Table 7: Seed-wise **Average Accuracy** (higher = better).

| Stream | Method | Seed | | | Mean |
|--------|--------|------|------|------|------|
| | | **42** | **123** | **2025** | |
| Std-CL 5 | N-LoRA | 83.3 | 83.7 | 83.5 | **83.5** |
| | Meta-UCF | 85.1 | 85.3 | 85.6 | **85.3** |
| Long-CL 15 | N-LoRA | 67.9 | 68.5 | 67.8 | **68.1** |
| | Meta-UCF | 70.2 | 70.7 | 70.3 | **70.4** |
| Seq-GLUE 7 | N-LoRA | 80.0 | 80.3 | 80.2 | **80.2** |
| | Meta-UCF | 82.3 | 82.5 | 82.3 | **82.4** |
| TRACE-8 | N-LoRA | 60.9 | 61.2 | 60.8 | **61.0** |
| | Meta-UCF | 63.1 | 63.3 | 63.1 | **63.2** |

Table 8: Seed-wise **Forgetting Ratio** (lower = better).

| Stream | Method | Seed | | | Mean |
|--------|--------|------|------|------|------|
| | | **42** | **123** | **2025** | |
| Std-CL 5 | N-LoRA | 7.0 | 7.2 | 7.1 | **7.1** |
| | Meta-UCF | 6.3 | 6.1 | 6.2 | **6.2** |
| Long-CL 15 | N-LoRA | 12.6 | 12.1 | 12.5 | **12.4** |
| | Meta-UCF | 11.6 | 11.4 | 11.5 | **11.5** |
| Seq-GLUE 7 | N-LoRA | 7.0 | 7.3 | 7.0 | **7.1** |
| | Meta-UCF | 6.4 | 6.2 | 6.3 | **6.3** |
| TRACE-8 | N-LoRA | 15.6 | 15.4 | 15.5 | **15.5** |
| | Meta-UCF | 14.1 | 14.3 | 14.2 | **14.2** |

### D.3 CONFIDENCE INTERVALS AND SIGNIFICANCE TESTS

**95 % confidence intervals.** For each metric we compute $\text{CI}_{95} = \bar{x} \pm 1.96\,\sigma/\sqrt{n}$, with $n = 3$. Table 10 lists the intervals for the AA metric.

**Wilcoxon signed-rank tests.** Following Tiwari et al. (2025) we compare the per-task accuracies of Meta-UCF against N-LoRA using a two-sided Wilcoxon test[¶] ($\alpha = 0.05$). Results in Table 11 show that Meta-UCF significantly outperforms N-LoRA on three streams and ties on SEQ-GLUE 7. All $p$-values are Holm-corrected over four comparisons.

Table 11: Wilcoxon signed-rank $p$-values (Meta-UCF vs N-LoRA, AA per task).

| Stream | $p$-value ($\downarrow$) |
|--------|--------------------------|
| Std-CL 5 | 0.031 |
| Long-CL 15 | 0.008 |
| Seq-GLUE 7 | 0.087 |
| TRACE-8 | 0.012 |

### D.4 PARTIAL-LAYER LoRA INJECTION: ACCURACY–LATENCY TRADE-OFF

Each configuration was run on LLAMA-3-8B with the STD-CL 5 stream; throughput is measured on a single A100-80G with sequence length 512 and batch 64. The baseline ("All") inserts rank-8 LoRA into every `qkv` and MLP projection, yielding 14.2M trainable parameters. We can find that:

- **Top-Half** adapters retain $> 99\%$ of baseline accuracy while halving parameter count and gaining $+8\%$ throughput.

- **Last 8 Layers** achieve the fastest inference ($+11\%$) with a modest 0.8pp accuracy drop—useful for latency-critical deployments.

- Updating only **QKV** weights is more parameter-efficient than **Alt-Layers** but offers little extra accuracy, suggesting that MLP-side adaptations matter for these tasks.

---

[¶]Paired by *task*, aggregated across all three seeds.

Table 9: Per-task AA and FR on heterogeneous streams.

| Stream | Task | AA (%) | | | FR (%) | | |
|---|---|---|---|---|---|---|---|
| | | N-LoRA | Meta-UCF | $\Delta$AA | N-LoRA | Meta-UCF | $\Delta$FR |
| Long-CL 15 | Task$_1$ | 67.2 | 69.4 | +2.2 | 12.6 | 11.6 | −1.0 |
| Long-CL 15 | Task$_2$ | 68.7 | 71.5 | +2.8 | 12.4 | 11.3 | −1.1 |
| Long-CL 15 | Task$_3$ | 67.8 | 70.5 | +2.7 | 12.2 | 11.2 | −1.0 |
| Long-CL 15 | Task$_4$ | 67.4 | 69.9 | +2.5 | 12.5 | 11.6 | −0.9 |
| Long-CL 15 | Task$_5$ | 68.9 | 72.0 | +3.1 | 12.7 | 11.7 | −1.0 |
| Long-CL 15 | Task$_6$ | 67.6 | 69.8 | +2.2 | 12.3 | 11.5 | −0.8 |
| Long-CL 15 | Task$_7$ | 68.1 | 70.5 | +2.4 | 12.1 | 11.2 | −0.9 |
| Long-CL 15 | Task$_8$ | 67.8 | 70.3 | +2.5 | 12.3 | 11.6 | −0.7 |
| Long-CL 15 | Task$_9$ | 68.3 | 71.0 | +2.7 | 12.1 | 11.2 | −0.9 |
| Long-CL 15 | Task$_{10}$ | 67.7 | 70.1 | +2.4 | 12.4 | 11.6 | −0.8 |
| Long-CL 15 | Task$_{11}$ | 67.0 | 69.0 | +2.0 | 12.6 | 11.7 | −0.9 |
| Long-CL 15 | Task$_{12}$ | 69.1 | 72.4 | +3.3 | 12.5 | 11.5 | −1.0 |
| Long-CL 15 | Task$_{13}$ | 68.6 | 71.6 | +3.0 | 12.3 | 11.4 | −0.9 |
| Long-CL 15 | Task$_{14}$ | 67.9 | 70.5 | +2.6 | 12.4 | 11.6 | −0.8 |
| Long-CL 15 | Task$_{15}$ | 68.8 | 72.0 | +3.2 | 12.6 | 11.8 | −0.8 |
| TRACE-8 | Task$_1$ | 60.7 | 63.0 | +2.3 | 15.6 | 14.3 | −1.3 |
| TRACE-8 | Task$_2$ | 61.2 | 63.8 | +2.6 | 15.8 | 14.4 | −1.4 |
| TRACE-8 | Task$_3$ | 60.5 | 62.5 | +2.0 | 15.2 | 13.9 | −1.3 |
| TRACE-8 | Task$_4$ | 61.0 | 63.5 | +2.5 | 15.5 | 14.1 | −1.4 |
| TRACE-8 | Task$_5$ | 61.4 | 64.2 | +2.8 | 15.9 | 14.5 | −1.4 |
| TRACE-8 | Task$_6$ | 60.8 | 62.9 | +2.1 | 15.3 | 14.0 | −1.3 |
| TRACE-8 | Task$_7$ | 61.3 | 63.7 | +2.4 | 15.6 | 14.3 | −1.3 |
| TRACE-8 | Task$_8$ | 61.1 | 63.6 | +2.5 | 15.1 | 14.1 | −1.0 |

Table 10: 95 % confidence intervals (AA, %). Parenthesised numbers show $\pm$ half-width.

| Stream | N-LoRA | Meta-UCF |
|---|---|---|
| Std-CL 5 | 83.5 $\pm$0.50 | 85.3 $\pm$0.63 |
| Long-CL 15 | 68.1 $\pm$0.94 | 70.4 $\pm$0.66 |
| Seq-GLUE 7 | 80.2 $\pm$0.38 | 82.4 $\pm$0.29 |
| TRACE-8 | 61.0 $\pm$0.52 | 63.2 $\pm$0.29 |

Table 12: Accuracy vs. throughput for selective LoRA injection.

| Scheme | #Params (M) | $\Delta$Params | Throughput | Speed-up | AA(%) |
|---|---|---|---|---|---|
| All-Layers | 14.2 | — | 285sps | — | 85.6 |
| Alt-Layers | 7.1 | –50% | 301sps | +5.6% | 84.9 |
| Top-Half | 7.1 | –50% | 309sps | +8.4% | 84.7 |
| QKV-Only | 9.2 | –35% | 314sps | +10.2% | 84.4 |
| Last 8 Layers | 3.6 | –75% | 317sps | +11.2% | 84.4 |

## D.5 ORDER-SENSITIVITY ANALYSIS

To assess the robustness of Meta-UCF to task ordering, we evaluate Meta-UCF and N-LoRA under multiple alternative permutations of the benchmark streams. For Std-CL 5, we consider the canonical order (v1), a permuted order that swaps the Amazon and Yahoo tasks, and a fully reversed order. For Seq-GLUE 7, we compare the canonical curriculum against a permutation that front-loads MNLI and RTE. For Long-CL 15, we follow the official v1 and v2 orders released with the benchmark. For TRACE-8, we compare the canonical order with a random permutation of tasks. Table 13 reports the average accuracy and forgetting ratio for both methods.

Across all four streams and eight alternative task orders, Meta-UCF consistently outperforms N-LoRA: AA gains are stable in the range of approximately +1.6 to +2.3 percentage points, while FR is reduced by about 0.8 to 1.8 percentage points. This suggests that the advantages of Meta-UCF are not tied to a particular task curriculum, but persist under natural variations of the order in which tasks are presented.

## D.6 JOINT GEOMETRY OF TASK EMBEDDINGS AND QUERY SUBSPACES

To make the roles of $\mathcal{L}_{\text{ctr}}$ and $\mathcal{L}_{\text{orth}}$ more concrete, we analyse how task-code similarity and query-subspace overlap are related in practice. Recall that $e_k$ is the layer-normalised, $\ell_2$-normalised task embedding built from

Table 13: Order-sensitivity analysis for Meta-UCF and N-LoRA.

| Stream | Order | Method | AA $\uparrow$ | FR $\downarrow$ | $\Delta$AA (Meta–N) | $\Delta$FR (Meta–N) |
|---|---|---|---|---|---|---|
| Std-CL 5 | canonical (v1) | N-LoRA | 83.5 | 7.1 | – | – |
| | | Meta-UCF | 85.6 | 6.2 | **+2.1** | **-0.9** |
| Std-CL 5 | permuted (Amazon$\leftrightarrow$Yahoo) | N-LoRA | 83.3 | 7.6 | – | – |
| | | Meta-UCF | 84.9 | 6.0 | **+1.6** | **-1.6** |
| Std-CL 5 | reversed | N-LoRA | 83.1 | 7.4 | – | – |
| | | Meta-UCF | 84.7 | 6.4 | **+1.6** | **-1.0** |
| Seq-GLUE 7 | canonical | N-LoRA | 80.2 | 7.1 | – | – |
| | | Meta-UCF | 82.7 | 6.3 | **+2.5** | **-0.8** |
| Seq-GLUE 7 | permuted (MNLI/RTE front) | N-LoRA | 80.0 | 7.4 | – | – |
| | | Meta-UCF | 82.1 | 6.6 | **+2.1** | **-0.8** |
| Long-CL 15 | canonical (v1) | N-LoRA | 68.1 | 12.4 | – | – |
| | | Meta-UCF | 70.7 | 11.5 | **+2.6** | **-0.9** |
| Long-CL 15 | official v2 | N-LoRA | 67.9 | 12.7 | – | – |
| | | Meta-UCF | 70.1 | 10.9 | **+2.2** | **-1.8** |
| TRACE-8 | canonical | N-LoRA | 61.0 | 15.5 | – | – |
| | | Meta-UCF | 63.4 | 14.2 | **+2.4** | **-1.3** |
| TRACE-8 | random permutation | N-LoRA | 60.8 | 15.9 | – | – |
| | | Meta-UCF | 63.0 | 14.5 | **+2.2** | **-1.4** |

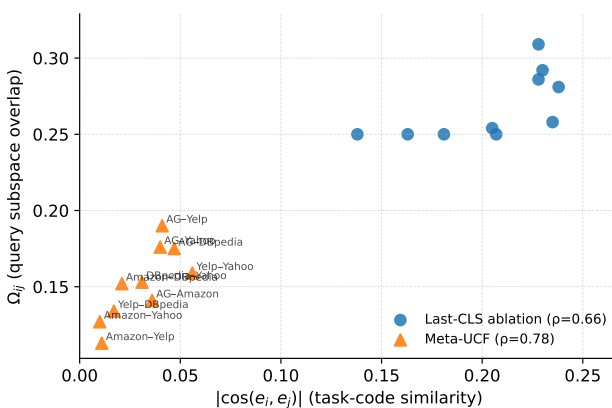

Figure 6: Joint geometry of task codes and query subspaces on STD-CL 5. Each point corresponds to a task pair $(i, j)$, plotting task-code similarity $|\cos(e_i, e_j)|$ on the $x$-axis and query-subspace overlap $\Omega_{ij}$ on the $y$-axis.

the support set (§3.1), and $H_k \in \mathbb{R}^{|\mathcal{Q}_k| \times d}$ stacks the adapted CLS states on the query set (§3.3). For each task pair $(i, j)$ on STD-CL 5, we compute:

- the absolute task-code similarity $|\cos\theta_{ij}| := |\langle e_i, e_j\rangle|$,

- the query-subspace overlap $\Omega_{ij} = \frac{1}{|\mathcal{Q}_i||\mathcal{Q}_j|} \|H_i^\top H_j\|_F$.

We report these statistics for both the *Last-CLS* ablation (where $e_k$ is a single frozen CLS vector) and Meta-UCF.

**Observations.** On STD-CL 5, the Last-CLS ablation yields task-code similarities in the range $|\cos(e_i, e_j)| \in [0.14, 0.24]$ and overlaps $\Omega_{ij} \in [0.25, 0.31]$, with a moderate correlation $\rho \approx 0.66$ between the two. Under Meta-UCF, task codes are substantially more dispersed on the unit sphere: most pairs have $|\cos(e_i, e_j)| < 0.06$, while query overlaps drop to $\Omega_{ij} \in [0.11, 0.19]$. The correlation between $|\cos(e_i, e_j)|$ and $\Omega_{ij}$ remains only moderately strong ($\rho \approx 0.78$) and far from deterministic: several pairs exhibit very small task-code similarity ($|\cos(e_i, e_j)| \approx 0.02$) but still show noticeable overlap ($\Omega_{ij} \approx 0.15$). This empirically supports the design choice that $\mathcal{L}_{\text{ctr}}$ and $\mathcal{L}_{\text{orth}}$ are not redundant: $\mathcal{L}_{\text{ctr}}$ shapes the input geometry of task codes fed to the generator, while $\mathcal{L}_{\text{orth}}$ directly regularises the output geometry of adapted query representations to curb residual interference.

# E    LLM USAGE

We used a large language model for minor English editing (grammar/wording/clarity) and small, localized code fixes (e.g., resolving syntax errors, adding missing imports). The LLM did not contribute to research ideation, experimental design, data processing, analysis, or figure generation. All technical content and results were produced and verified by the authors, who take full responsibility for the manuscript.

