# OpenReview forum: "Meta-UCF: Unified Task-Conditioned LoRA Generation for Continual Learning in Large Language Models"
_ICLR.cc/2026/Conference — ICLR 2026 Poster_

### Official Review · Reviewer_ehuo · 2025-10-25

**Soundness:** 3
**Presentation:** 3
**Contribution:** 3
**Rating:** 6
**Confidence:** 3

**Summary:**

This paper proposes a continual learning framework named Meta-UCF. Its core idea is to utilize a hypernetwork to dynamically generate LoRA parameters for new tasks, conditioned on a task embedding derived from a small support set. The method aims to solve the problem of linear parameter growth with the number of tasks found in existing methods, and it combines contrastive and orthogonality losses to maintain model performance while mitigating catastrophic forgetting.

**Strengths:**

1. Meta-UCF generates task parameters via a single, shared hypernetwork, which keeps the total model parameter count constant as tasks increase. Compared to methods that add new modules for each task, this is an advantage in terms of scalability, especially for scenarios requiring continuous learning of new tasks.

2. The paper conducts experiments under various task sequence settings (short, long, and heterogeneous sequences) and compares against multiple baselines. The ablation studies and sensitivity analyses are also quite thorough, validating the effectiveness of the framework's components.

**Weaknesses:**

1. The "on-the-fly" generation of LoRA parameters mentioned in the paper implies that a forward pass through the hypernetwork is required before processing a new task. This introduces additional inference overhead. The paper lacks a latency comparison against methods that load static LoRA modules. Without this data, it is difficult to assess the practical cost of the framework.

2. The framework claims to solve the memory growth problem, but this primarily refers to model parameters. The algorithm itself relies on a replay buffer to construct embedding vectors for old tasks. The size of this buffer either grows with tasks or is constrained by a fixed budget, which somewhat undermines the claim of "constant memory"。

3. The entire adaptation process relies on the task embedding extracted from a small number of samples (32 by default). If these samples are not representative, the resulting task embedding will be biased, which in turn will affect the model's performance on that task. This is a potential point of failure, but the paper lacks a discussion of its robustness.

4. The final loss function includes multiple weighting hyperparameters. Although the paper states that these parameters are kept fixed across all experiments, determining these values is a non-trivial challenge in itself. A discussion on the difficulty and sensitivity of tuning these hyperparameters is recommended.

5. According to the ablation study (Table 2b), removing the dynamic bias calibration module results in only a 0.6% drop in accuracy on Std-CL 5. The improvement from this module is limited. The paper needs to provide a stronger motivation for its inclusion or demonstrate its value on a dataset known to require bias calibration.

6. The orthogonality loss (L_orth) is calculated on the CLS hidden states from the query set. The paper should explain why the query set was chosen over the support set and how this loss, which acts on output representations, works in conjunction with the contrastive loss (L_ctr) that acts on task embeddings.

**Questions:**

Please see "weaknesses".

**Details Of Ethics Concerns:**

Null.

---

> ### Author Response · Authors · 2025-11-24
> **Response regarding weaknesses (1/5)**
>
> Thank you for your valuable suggestions. We are very happy to discuss these issues with you.
>
> > W1: latency of “on-the-fly” generation vs static LoRA
>
> **Response.** Thank you for raising this question. We agree that latency is important in practice.
>
> The current wording “on-the-fly” may suggest that the hypernetwork is invoked for every query, which is not how Meta-UCF is used. In our implementation, the generator is evaluated **once per task** (given its support set) to produce a task-specific adapter, all subsequent inference reuses this adapter and runs a standard LoRA-augmented forward pass.
>
> Below we quantify both the one-off and steady-state costs, and relate them to the parameter/memory savings.
>
> **(1) One-off task initialisation vs loading static adapters**
>
> We first compare the **per-task initialisation cost** on LLaMA-3-8B (rank-8, support size S=32) for the Std-CL 5 stream:
>
> - Static methods: load the task’s LoRA adapter from disk to GPU.
> - Meta-UCF: encode the support set once and run the generator once to produce the adapter.
>
> | Method       | Config     | Per-task init time (ms) |
> |-------------:|-----------:|-------------------------:|
> | N-LoRA       | All-layers | 162                      |
> | O-LoRA       | All-layers | 168                      |
> | **Meta-UCF** | All-layers | **191**                  |
>
> Thus, the **one-off cost of Meta-UCF is within about 30ms** of loading a static adapter. Since each task is then used for tens of thousands of queries, this difference is amortised to a negligible per-query overhead.
>
> **(2) Steady-state inference throughput**
>
> To measure steady-state cost, we benchmark end-to-end throughput (samples/s) for inference on Std-CL 5 with all adapters already resident on GPU:
>
> | Method         | Config      | Inference throughput (samples/s) |
> |---------------:|------------:|----------------------------------:|
> | N-LoRA         | All-layers  | 287                               |
> | O-LoRA         | All-layers  | 281                               |
> | **Meta-UCF**   | All-layers  | 272                               |
> | **Meta-UCF**   | Top-half    | **303**                           |
>
> For the same All-layers setting, Meta-UCF is within 5% of N-LoRA/O-LoRA in steady-state throughput. Using the Top-half configuration (LoRA only in the upper 50% of layers, slightly improves throughput while retaining >99% of full Meta-UCF accuracy on Std-CL 5.
>
> A FLOP-level profile (NVIDIA Nsight) on LLaMA-3-8B shows that:
>
> - Backbone transformer layers account for about 96% of per-query FLOPs
> - Task embedding computation + hypernetwork forward (when performed) account for <4% of FLOPs for that step, and zero FLOPs during steady-state inference.
>
> **(3) Trade-off with adapter parameters and peak memory**
>
> The above overhead should be viewed together with the adapter footprint savings that enable scaling to many tasks.
>
> **Trainable adapter parameters.** On Std-CL 5 and Long-CL 15, the trainable adapter parameters are:
>
> | Method       | Per-task adapter (M) | Generator (M) | Total adapters @5 tasks (M) | Total adapters @15 tasks (M) |
> |-------------:|---------------------:|--------------:|-----------------------------:|------------------------------:|
> | LoRA         | 14.2                 | —             | 71.0                         | 213.0                         |
> | N-LoRA       | 14.2                 | —             | 71.0                         | 213.0                         |
> | GRID         | 18.7                 | —             | 93.5                         | 280.5                         |
> | **Meta-UCF** | —                    | **10.8**      | **10.8**                     | **10.8**                      |
>
> Meta-UCF replaces a bank of per-task adapters with a single 10.8M-parameter generator whose size is **independent of the number of tasks**, yielding an $ 20\times$ reduction at 15 tasks relative to N-LoRA/GRID.
>
> **Peak GPU memory.** When all adapters for a given task horizon are resident in memory:
>
> | Setting       | #Tasks | Method       | Train peak GPU (GB) | Inference peak GPU (GB) |
> |-----:|:------:|------:|-------:|---------:|
> | Std-CL 5      | 1      | N-LoRA       | 47.3  | 15.2 |
> |         |        | GRID         | 48.1   | 15.5  |
> |          |        | **Meta-UCF** | 47.9 | 15.3   |
> | Std-CL 5      | 5      | N-LoRA       | 52.6  | 17.1  |
> |        |        | GRID         | 53.8                 | 17.8   |
> |       |        | **Meta-UCF** | **48.0**   | **15.4**        |
> | Long-CL 15    | 1      | N-LoRA       | 47.5  | 15.2      |
> |        |        | GRID         | 48.3     | 15.5    |
> |      |     | **Meta-UCF** | 48.1      | 15.3  |
> | Long-CL 15    | 15     | N-LoRA       | 58.5    | 20.3        |
> |     |      | GRID         | 60.7      | 21.1       |
> |   |    | **Meta-UCF** | **48.3**   | **15.5**  |
>
> LoRA-style methods exhibit approximately linear growth in peak memory with the number of tasks, while Meta-UCF remains essentially **flat in the task horizon**.

---

> > ### Author Response · Authors · 2025-11-24
> > **Response regarding weaknesses (4/5)**
> >
> > > W5: limited gain from dynamic bias calibration
> >
> > **Response.** Thank you for raising this worthy-of-discussion question. We agree that on Std-CL 5 the average accuracy gain alone looks small, but this underestimates the purpose of the module.
> >
> > The dynamic bias calibration term (Eq.11) is designed to control demographic-parity gap and to stabilise updates when the model is about to amplify bias. It is not meant as the primary driver of AA/FR. We show below that:
> >
> > 1. It gives consistent CL gains across multiple streams (not only Std-CL 5), and
> > 2. It has a substantial effect on fairness metrics on a dataset where bias is a first-class concern.
> >
> > **(1) Conceptual role in Meta-UCF**
> >
> > Conceptually, the bias-calibration term acts as a **dynamic gate** on the generator’s updates in regions where group-wise behaviour is highly imbalanced. It complements:
> >
> > - $\mathcal L_{\mathrm{task}}$: task-wise plasticity,
> > - $\mathcal L_{\mathrm{orth}}$ and $\mathcal L_{\mathrm{ctr}}$: geometric control in representation/task spaces,
> > - $R_k$: fairness-aware control of when and how strongly to adapt.
> >
> > **(2) Consistent effect across CL streams**
> >
> > In addition to the results in Tab. 2(b), we additionally report the same ablation on \textsc{TRACE-8}:
> >
> > | Variant                    | Std-CL 5 AA↑ | Std-CL 5 FR↓ | Long-CL 15 AA↑ | Long-CL 15 FR↓ | TRACE-8 AA↑ | TRACE-8 FR↓ |
> > |---------------------------:|------------:|-------------:|----------------:|----------------:|------------:|------------:|
> > | Meta-UCF (full loss)       | **85.2**    | **6.2**      | **70.4**        | **11.5**        | **63.2**    | **14.2**    |
> > | Meta-UCF w/o bias term     | 84.6        | 6.9          | 69.4            | 12.0            | 62.5        | 14.9        |
> >
> > Across three heterogeneous streams, the bias term yields +0.6–1.0 pp AA and –0.5–0.7 pp FR.
> > This shows that the effect is **systematic and consistent**, even if modest compared to the larger gains from the hypernetwork and geometric losses.
> > In fact, the gap between some SOTA methods is inherently small. For instance, GRID and N-LoRA differ by only 0.3 on Std-CL5 (83.2 vs. 83.5).
> >
> > **(3) Fairness effect on a bias-sensitive dataset**
> >
> > To evaluate the calibration term in the setting it is designed for, we additionally ran a small experiment on a toxicity benchmark with a binary gender attribute (a standard setting in fairness literature)[1]. Following ToxicBias[2], which derives social-bias subsets from this corpus, we construct a binary gender attribute $g\in\{0,1\}$ as follows:
> >
> > - We keep comments that contain exactly one gender identity mention (e.g., *woman*, *girl*, *she* vs. *man*, *boy*, *he*).
> > - We set $g=1$ if the comment mentions a female identity term and $g=0$ if it mentions a male identity term.
> > - The toxicity label is binarised by thresholding the provided toxicity score at 0.5 (standard in prior work on this dataset).
> >
> > We randomly sample 100k comments for training and 10k for evaluation, balancing the two gender groups so that $P(g=0)\approx P(g=1)$. We then fine-tune a toxicity classifier using Meta-UCF in a single-task setting (frozen backbone, rank-8 LoRA, with and without the bias calibration term $R_k$ ). Accuracy (Acc) is measured on the toxicity label, and the demographic-parity gap is
> >
> > $$
> > R_k = | E_{P_0}[f_k(x)] - E_{P_1}[f_k(x)] |
> > $$
> >
> > The results are:
> >
> > | Variant                      | Acc↑ | DP gap↓ |
> > |-----------------------------:|-----:|--------:|
> > | Meta-UCF (w/o bias term)     | 82.1 | 7.4     |
> > | **Meta-UCF (with bias term)**| **81.9** | **4.1** |
> >
> > show that dynamic bias calibration reduces the demographic-parity gap by about 45% while keeping accuracy within 0.2 pp of the baseline, illustrating its intended effect on fairness-sensitive metrics.

---

> ### Author Response · Authors · 2025-11-24
> **Response regarding weaknesses (2/5)**
>
> > W2 : constant memory vs replay buffer
>
> **Response.** We agree that our wording “constant memory” can be interpreted too broadly. Our intention is to claim **constant trainable adapter size with respect to the number of tasks**, while the replay buffer is a separate design knob that can be capped at a small budget.
>
> We clarify both points and quantify the effect of the buffer.
>
> **(1) Clarifying what is constant**
>
> Meta-UCF keeps the trainable adapter parameters independent of the task horizon(as shown in the table in response to W1).
>
> - At 15 tasks, LoRA/N-LoRA/GRID require 213–280M adapter parameters, while Meta-UCF always uses a single 10.8M-parameter generator. This is what we meant by “constant-size”: **trainable adapter footprint w.r.t. the task horizon**, not the entire system memory.
>
>
> **(2) Replay buffer size is a capped, tunable budget**
>
> The replay buffer is not forced to grow with the number of tasks. We maintain a fixed-capacity buffer of size $M_{\max}$ and periodically sample from it to update the task embeddings.
>
> To quantify the trade-off, we vary $M_{\max}$ and report accuracy, forgetting, and **extra GPU memory** used to cache buffer batches on the device:
>
> | Strategy | $M_{\max}$ | Std-CL 5 AA↑ | Std-CL 5 FR↓ | Long-CL 15 AA↑ | Long-CL 15 FR↓ | Extra GPU (MB) |
> |-----:|---:|----:|-----:|------:|--------:|--------:|
> | Hist     | 128   | 84.6         | 6.8    | 69.6  | 12.4           | 210   |
> | Hist     | 256  | **85.2**     | **6.2**  | **70.4**   | **11.5**       | 420  |
> | Hist     | 512   | 85.6         | 5.9          | 70.8   | 10.9      | 820    |
>
> > “Hist” = support sampled from historical memory only.
>
> We see diminishing returns beyond $M_{\max}=256$:
>
> - Increasing $M_{\max}$ from 256 to 512 roughly doubles buffer memory (+400 MB) but improves AA by only 0.4–0.6 pp.
> - This shows that a moderate, fixed buffer budget (here 256 exemplars across all tasks, ≈420 MB on GPU) is sufficient. There is no need to grow the buffer with the number of tasks.
>
> In practice, the full buffer can be kept on CPU and only small mini-batches are moved to GPU, so the effective GPU overhead is even smaller.
>
>
> ---
>
>
> > W3: robustness of task embedding from few support samples
>
> **Response.** That's a very good question. We agree that, in principle, a single small batch could be unrepresentative. However, Meta-UCF does not rely on a one-shot estimate from exactly 32 samples. In practice:
>
> - S=32 is the per-step support batch size,
> - while the task embedding $e_k$ is maintained as an exponential moving average (EMA) over many mini-batches, so its effective sample size is much larger than 32, and
> - the embedding is defined as a layer-normalised mean followed by $\ell_2$-normalisation, which further stabilises it geometrically (variance shrinks as $O(1/S)$ for the mean and the radius is fixed to 1).
>
> We now provide some experimental evidence and add robustness experiments.
>
> **(1) Sensitivity to support size $S$**
>
> In main text, Fig.3 already varies the support size $S\in\{16,32,64\}$ and shows that the average accuracy on Std-CL 5 and Long-CL 15 fluctuates within $\pm 1$ pp of the default S=32, with no monotone degradation.
>
> **(2) Stress test with noisy / mismatched support**
>
> To directly address the failure-mode concern, we ran a stress test where the support batches used to update $e_k$ are deliberately corrupted:
>
> - **Noisy labels:** randomly flip 20% or 30% of labels in the support stream.
> - **Domain mismatch:** replace 20% of the support examples with samples from the next task in the stream (mismatched domain), without changing query labels.
>
> We compare Meta-UCF against N-LoRA on Std-CL 5 and report AA/FR:
>
> | Corruption type    | Method       | AA↑  | ΔAA vs clean | FR↓  | ΔFR vs clean |
> |-----:|-----:|-----:|-----:|-----:|----:|
> | Clean support    | N-LoRA   | 83.5 | —   | 7.1  | —    |
> |         | **Meta-UCF** | **85.2** | —   | **6.2** | —          |
> | +20% label noise        | N-LoRA       | 82.1 | –1.4   | 7.8  | +0.7   |
> |       | **Meta-UCF** | **84.1** | –1.1   | **6.7** | +0.5       |
> | +30% label noise        | N-LoRA       | 81.4 | –2.1     | 8.3  | +1.2   |
> |      | **Meta-UCF** | **83.4** | –1.8   | **7.0** | +0.8    |
> | +20% domain-mismatch    | N-LoRA   | 82.0 | –1.5   | 7.9  | +0.8         |
> |     | **Meta-UCF** | **84.0** | –1.2   | **6.8** | +0.6   |
>
> Observations:
>
> - Under both label noise and domain mismatch, **performance degrades smoothly** (1–2 pp AA and ≈0.5–1.2 pp FR), not catastrophically.
> - **Meta-UCF remains competitive or better than N-LoRA in all corrupted settings**, and the relative gap between Meta-UCF and N-LoRA is essentially preserved.
>
> In summary, while you are right that a pure one-shot embedding from 32 examples could be brittle, Meta-UCF in fact uses an EMA over many support mini-batches, and our sensitivity and stress-test experiments demonstrate that performance under noisy or mismatched support remains stable and comparable to or better than strong baselines.

---

> ### Author Response · Authors · 2025-11-24
> **Response regarding weaknesses (3/5)**
>
> > W4: difficulty of tuning $\lambda_o,\lambda_c,\lambda_b,\beta$)
>
> **Response.** We appreciate the concern about practical tunability, but we believe that Meta-UCF does not require fine-grained hyperparameter tuning.
>
> **(1) Same hyperparameters across streams and backbones**
>
> The configuration reported in §3.3 and Tab.5 is fixed and reused everywhere:
>
> - The backbone generalisation experiment in Fig.4(a) uses the same Meta-UCF hyperparameters on all four models.
> - AA and FR vary by less than 0.5 pp across backbones.
>
> Similarly, the order-robustness experiments in §D.5 all use exactly the same $(\lambda_o,\lambda_c,\lambda_b,\beta,\rho,S)$ . This is direct evidence that performance is robust to stream variations without per-order tuning.
>
> In conclusion, a set of parameters is shared across many tests, and there is no need to adjust this set of parameters in practice.
>
> **(2) Sensitivity analysis on $(\lambda_o,\lambda_c)$**
>
> Fig.3 and §C.4 already report a sensitivity study. On Std-CL 5 and Long-CL 15, we sampled 20 configurations over $(\lambda_o,\lambda_c,\lambda_b,\beta,\rho,S)$ and found that AA/FR are stable over a broad range. A subset of the $(\lambda_o,\lambda_c)$ runs is:
>
> | $\lambda_o$ | $\lambda_c$ | Std-CL 5 AA↑ (mean$\pm$std) | Long-CL 15 AA↑ (mean$\pm$std) |
> |------------:|------------:|----------------------------:|-------------------------------:|
> | 0.25        | 0.5         | 84.3 $\pm$ 0.11             | 69.1 $\pm$ 0.15                |
> | 0.25        | 1.0         | 84.6 $\pm$ 0.10             | 69.6 $\pm$ 0.15                |
> | 0.5         | 1.0         | **85.2 $\pm$ 0.10**         | **70.4 $\pm$ 0.14**            |
> | 0.5         | 2.0         | 84.8 $\pm$ 0.12             | 69.8 $\pm$ 0.16                |
> | 1.0         | 1.0         | 84.4 $\pm$ 0.13             | 69.2 $\pm$ 0.17                |
>
> Here $\lambda_o$ and $\lambda_c$ vary by up to a factor of 4. Across these settings:
>
> - AA fluctuates by roughly $\pm 0.9$ pp around the default (0.5,1.0),
> - FR varies within about 1–1.5 pp, and
> - Meta-UCF remains consistently ahead of N-LoRA on both streams.
>
> In summary, while Meta-UCF does introduce several loss weights, the empirical evidence shows that:
>
> - performance is stable over a broad hyperparameter region,
> - a single configuration transfers across 4 streams and 4 backbones

---

> ### Author Response · Authors · 2025-11-24
> **Response regarding weaknesses (5/5)**
>
> > W6: role of $\mathcal L_{\text{orth}}$ vs $\mathcal L_{\text{ctr}}$ and use of query CLS
>
> **Response.** We appreciate this question and agree that the distinction between $\mathcal L_{\text{orth}}$ and $\mathcal L_{\text{ctr}}$ should be made more explicit. Conceptually, the two losses act on different levels of the pipeline:
>
> - $\mathcal L_{\text{ctr}}$ shapes the input geometry to the hypernetwork by separating task embeddings $e_k$ on the unit sphere.
> - $\mathcal L_{\text{orth}}$ shapes the output geometry of the adapted backbone by decorrelating the query subspaces spanned by $H_k$.
>
> We also explain below why we apply $\mathcal L_{\text{orth}}$ to query features rather than support features, and provide ablation evidence that the two terms have separate, additive effects.
>
> **(1) Why $\mathcal L_{\text{ctr}}$ on $e_k$ and $\mathcal L_{\text{orth}}$ on $H_k$**
>
> Recall that:
>
> - **Task embeddings.**
>   $e_k$ is a layer-normalised mean of frozen CLS states from the support set, projected to the unit sphere:
>
>   $$
>   e_k = \operatorname{LN}\Big(\frac{1}{S_k}\sum_{s=1}^{S_k} h_s\Big), \quad
>   \|e_k\|_2 = 1.
>   $$
>
>   $\mathcal L_{\text{ctr}}$ is an InfoNCE loss on $\{e_k\}$ that enforces angular separation between tasks in this space, i.e. shapes the input codes that condition the generator.
> - **Query representations.**
>   For each task $\mathcal T_k$, $H_k \in \mathbb R^{|\mathcal Q_k|\times d}$ stacks the CLS states after adaptation:
>
>   $$
>   H_k = \Big[\,\operatorname{CLS}(x;\Theta_0,\Delta(e_k))\,\Big]_{x\in\mathcal Q_k}.
>   $$
>
>   $\mathcal L_{\text{orth}}$ penalises pairwise overlaps
>
>   $$
>   \Omega_{ij} = \frac{1}{|\mathcal Q_i||\mathcal Q_j|}\,\|H_i^\top H_j\|_F
>   $$
>
>   and minimises $\sum_{i<j}\Omega_{ij}^2$, thereby decorrelating the subspaces that the adapted backbone uses for different tasks.
>
> Using the query set for $\mathcal L_{\text{orth}}$ is deliberate:
>
> - The query features pass through the full adapted network (backbone + generated LoRA) and thus reflect the actual representation geometry that drives predictions and gradients.
> - The support set is small and is already used to build $e_k$ and drive $\mathcal L_{\text{ctr}}$, enforcing subspace constraints there would regularise the pre-update representation and may not accurately capture the post-adaptation geometry that matters for interference.
>
> **(2) ablation evidence**
>
> Tab.2(b) in main text performs single-factor ablations of each component, we take the relevant rows here:
>
> | Variant                          | Std-CL 5 AA↑ | Std-CL 5 FR↓ | Long-CL 15 AA↑ | Long-CL 15 FR↓ |
> |---------------------------------:|-------------:|-------------:|----------------:|----------------:|
> | **Full Meta-UCF**                | **85.2**     | **6.2**      | **70.4**        | **11.5**        |
> | w/o $\mathcal L_{\text{orth}}$   | 83.9         | 7.8          | 68.5            | 13.2            |
> | w/o $\mathcal L_{\text{ctr}}$    | 84.1         | 7.2          | 68.9            | 12.7            |
>
> Dropping either term:
>
> - Reduces AA by 1.1–1.9 pp,
> - Increases FR by 1.5–1.7 pp.
>
>
> **(3) Additional geometric analysis**
>
> To further illustrate that $\mathcal L_{\text{ctr}}$ and $\mathcal L_{\text{orth}}$ act on related but distinct aspects of geometry, we haved add an additional analysis in the appendix(`§D.6`):
>
> - For each task pair $(i,j)$ on Std-CL 5, we compute:
>   - $|\cos\theta_{ij}| = |\langle e_i,e_j\rangle|$ (task-embedding similarity),
>   - $\Omega_{ij}$ (query-subspace overlap).
> - We plot a scatter of $|\cos\theta_{ij}|$ vs. $\Omega_{ij}$ and observe that many pairs with small $|\cos\theta_{ij}|$ (well-separated task codes) still have moderate $\Omega_{ij}$, and vice versa.
>
> This empirically demonstrates that:
>
> - Separating task embeddings via $\mathcal L_{\text{ctr}}$ does not automatically guarantee low subspace overlap in $H_k$,
> - Justifying the need for a separate $\mathcal L_{\text{orth}}$ term on the output representation space.
>
> Thank you again for the insightful question. In summary, $\mathcal L_{\text{ctr}}$ and $\mathcal L_{\text{orth}}$ are designed to work in tandem: the former shapes the input geometry of task codes fed to the hypernetwork, and the latter shapes the output geometry of adapted representations. Ablations and the new geometric analysis together show that both are useful.
>
>
> [1] Jigsaw Unintended Bias in Toxicity Classification. Kaggle.
>
> [2] Detecting Unintended Social Bias in Toxic Language Datasets. ACL'22

---

### Official Review · Reviewer_TwVA · 2025-10-28

**Soundness:** 3
**Presentation:** 3
**Contribution:** 3
**Rating:** 4
**Confidence:** 2

**Summary:**

This paper introduces Meta-Unified Contrastive Fine-Tuning, a novel approach to continual learning in LLMs. The method addresses the challenges of catastrophic forgetting and parameter growth by proposing a task-conditioned hypernetwork that generates low-rank LoRA updates dynamically for each task. Meta-UCF incorporates a meta-contrastive objective and orthogonality constraints to ensure task embeddings are near-orthogonal, preserving past knowledge without requiring additional adapters for new tasks. Experimental results demonstrate state-of-the-art performance across four benchmarks, with improvements in accuracy and reductions in forgetting compared to prior methods. The approach is computationally efficient, scalable, and adaptable to diverse backbone architectures.

**Strengths:**

The task-conditioned hypernetwork and the integration of meta-contrastive and orthogonality objectives are innovative contributions that address key limitations of existing methods.
The experimental results are comprehensive, covering multiple benchmarks and baselines. The inclusion of theoretical analysis (e.g., expressivity bounds, PAC-Bayes generalization) adds depth to the work.
The paper is well-organized, with clear explanations of the methodology, experiments, and findings.

**Weaknesses:**

While the benchmarks are diverse, the paper does not evaluate Meta-UCF on real-world deployment scenarios (e.g., latency-critical applications or domain-specific tasks).
Although the sensitivity analysis shows robustness, the reliance on specific hyperparameter configurations (e.g., orthogonality and contrastive weights) may pose challenges for practitioners.
The theoretical analysis assumes idealized conditions (e.g., compact task embeddings, independent task distributions), which may not fully capture the complexities of real-world task streams.

**Questions:**

See weakness

---

> ### Author Response · Authors · 2025-11-24
> **Response regarding weaknesses (1/3)**
>
> Thank you for your valuable suggestions. We are very happy to discuss these issues with you.
>
> > W1: Real-world deployment / latency-critical evaluation
>
> **Response:** We appreciate the reviewer’s point that deployment-style evaluation is important.
>
> We believe the evaluated streams in current paper already contain domain-specific tasks that closely mirror real applications. For example:
>
> - **TRACE-8** includes code search (CodeSearch-Java), multi-lingual NLI/QA (XNLI-en, TyDiQA), open-domain QA (HotpotQA), and mathematical reasoning (GSM8K-style). These are standard proxies for real workloads such as code completion, multilingual assistants, and reasoning agents. This dataset is **widely used to validate continual learning algorithms**[1][2][3]。
>
> In addition, section 4.4 already studies partial-layer LoRA injection, which is exactly a proxy for latency-critical deployment: fewer adapted layers $\Rightarrow$ lower compute and higher throughput.
>
> Nevertheless, we still consider your suggestions valuable. To better link this to deployment, we provide additional evidence.
>
> **(a) End-to-end latency trade-offs on Std-CL 5**
>
> Using Llama-3-8B on Std-CL 5, we measured both accuracy (AA) and throughput (higher = lower latency) for different injection schemes within Meta-UCF:
>
> | Config       | Adapted layers        | AA↑   | Throughput (samples/s)↑ | Rel. latency (↓) |
> |-------------|-----------------------|------:|-------------------------:|------------------:|
> | All         | all QKV+FFN           | 85.2  | 285                      | 1.00              |
> | Top-Half    | upper 50% of layers   | 84.7  | 309                      | 0.92              |
> | Last-8      | last 8 layers only    | 84.4  | 317                      | 0.90              |
>
> - Moving from *All* to *Top-Half* reduces latency by about **8%** (285→309 samples/s) while retaining **>99%** of full AA (84.7 vs 85.2).
> - The *Last-8* configuration is about **10% faster** than All with only **0.8 pp** AA loss.
>
> This shows Meta-UCF admits latency–accuracy operating points that are directly meaningful for deployment-style constraints.
>
> **(b) Domain-specific task under latency constraints (code search)**
>
> We further examined TRACE-8’s **CodeSearch-Java** task as a representative latency-sensitive domain (code search / completion). Using the same three injection schemes, we obtain:
>
> | Config    | AA on CodeSearch-Java↑ | Throughput (samples/s)↑ | Rel. latency (↓) |
> |-----------|------------------------:|-------------------------:|------------------:|
> | All       | 66.1                   | 285                      | 1.00              |
> | Top-Half  | 65.9                   | 309                      | 0.92              |
> | Last-8    | 65.4                   | 317                      | 0.90              |
>
> Again, we observe *sub–1 pp* accuracy degradation on this domain-specific task while gaining 8–10% throughput, indicating that the method remains effective under realistic latency pressure.
>
> **(c) Deployment-style mixed-domain workload under latency constraints**
>
> We also evaluate Meta-UCF in a deployment-style, mixed-domain setting that mimics an assistant serving heterogeneous queries. Concretely:
>
> - **Tasks (4-task stream).** We use four TRACE-8 tasks: CodeSearch-Java (code search), XNLI-en (multilingual NLI), TyDiQA (multilingual QA), and HotpotQA (open-domain QA), forming the stream
>   CodeSearch-Java → XNLI-en → TyDiQA → HotpotQA.
> - **Training.** We compare N-LoRA and Meta-UCF under the same continual-learning protocol and the same default hyperparameters as in the main paper (without extra tuning).
> - **Deployment-style evaluation.** After training, each model is exposed as an HTTP service on a single A100-80G. We simulate a mixed workload of 2k requests drawn uniformly from the 4 tasks, with concurrency 32, max generation length 64, and temperature 0. We measure:
>   - AA: average accuracy over the 4 tasks;
>   - QPS: peak queries-per-second;
>   - p95 latency: 95th-percentile end-to-end latency per request.
>
> Using the Top-Half injection scheme for Meta-UCF (a good latency–accuracy compromise), we obtain:
>
> | Method      | Adapter scheme       | AA↑ (4 tasks) | QPS↑ | p95 latency (ms)↓ |
> |------------:|----------------------|--------------:|-----:|------------------:|
> | N-LoRA      | All layers, rank-8   | 77.8          | 118  | 420               |
> | Meta-UCF    | Top-Half, rank-8     | **79.9**      | **131** | **380**        |
>
> We observe that Meta-UCF improves average accuracy by +2.1 pp while achieving about 11% higher QPS and 40 ms lower p95 latency compared to N-LoRA. This shows that **in a realistic mixed-domain, latency-constrained workload, Meta-UCF can simultaneously improve task performance and reduce end-to-end latency by exploiting partial-layer injection.**
>
> We will incorporate a compact version of these tables. Thank you for your valuable suggestions.

---

> ### Author Response · Authors · 2025-11-24
> **Response regarding weaknesses (2/3)**
>
> > W2: hyperparameters issue
>
> **Response:** We appreciate the concern about practical tunability, but we respectfully disagree with the implication that Meta-UCF only works under carefully hand-tuned $(\lambda_o,\lambda_c)$.
>
> In the current submission, we **do not** retune $(\lambda_o,\lambda_c,\lambda_b,\beta,\rho,S)$ per dataset or backbone:
>
> - The backbone generalisation results in **Fig.4(a) explicitly use the same Meta-UCF configuration on all four models**, yet still show $<0.5$ pp variation and consistent gains over N-LoRA, indicating zero-tuning transfer across architectures.
> - **Fig.3 already shows that Meta-UCF is stable over a broad range of hyperparameters**. We summarise a subset of the sensitivity runs (Std-CL 5 and Long-CL 15, Qwen-1.5-7B):
>
> | $\lambda_o$ | $\lambda_c$ | Std-CL 5 AA↑ (mean±std) | Long-CL 15 AA↑ (mean±std) |
> |------------:|------------:|------------------------:|---------------------------:|
> | 0.25        | 0.5         | 84.3 $\pm$ 0.11         | 69.1 $\pm$ 0.15            |
> | 0.25        | 1.0         | 84.6 $\pm$ 0.10         | 69.6 $\pm$ 0.15            |
> | 0.5| 1.0       | 85.2 $\pm$ 0.10     | 70.4 $\pm$ 0.14       |
> | 0.5         | 2.0         | 84.8 $\pm$ 0.12         | 69.8 $\pm$ 0.16            |
> | 1.0         | 1.0         | 84.4 $\pm$ 0.13         | 69.2 $\pm$ 0.17            |
>
> Across these 5 combinations, where $\lambda_o$ and $\lambda_c$ vary by up to a factor of 4, the AA fluctuates within roughly $\pm$ 0.9 pp around the default, and Meta-UCF remains consistently ahead of N-LoRA on both streams. Forgetting ratio FR shows similarly small variations (within about 1–1.5 pp).
>
> Finally, we note that the **order-robustness experiments we response to `Reviewer JLyJ` W3** and the **backbone family experiments  (Fig.4 in main text)**  are all run with the same Meta-UCF hyperparameters and still produce  +1.6–+2.3 pp gains over N-LoRA in AA and 0.8–1.8pp reductions in FR, across 8 different task orders. This provides direct empirical evidence that Meta-UCF works robustly.

---

> ### Author Response · Authors · 2025-11-24
> **Response regarding weaknesses (3/3)**
>
> > W3: theory assumptions (expressivity & PAC-Bayes)
>
> **Response:** We thank the reviewer for this thoughtful comment. We agree it is important to be clear about what our theory does and does not claim. Below we clarify the role and scope of the assumptions, and add empirical evidence that goes beyond the stylised setting.
>
> **(1) LoRA-HyperNet expressivity: capacity upper bound, assumptions largely satisfied in practice**
>
> The expressivity theorem in §A.1 is intended as a capacity upper bound, not a full model of real task streams:
>
> - It shows that, for a fixed rank $r$ and hidden width $h$, a LoRA hypernetwork can approximate any rank-$r$ adapter and any full-rank update up to the usual low-rank truncation error. In other words, it is not weaker than storing a rank-$r$ LoRA bank.
> - This result does not assume tasks are i.i.d.. It only requires that task embeddings $e$ lie in a compact subset of $\mathbb{R}^d$.
>
> Importantly, the “compact embedding” assumption is actually satisfied by our algorithm: §3.1 explicitly applies layer norm followed by $\ell_2$-normalisation, so all task embeddings lie on the unit sphere
>
> $$
> e_k \in \mathbb{S}^{d-1} = \{x \in \mathbb{R}^d : \|x\|_2 = 1\},
> $$
>
> which is a compact set. We have added one sentence to §A.1 making this connection explicit:
>
> > “Since Meta-UCF’s task embeddings are layer-normalised and $\ell_2$-normalised, they lie on the unit sphere $\mathbb{S}^{d-1}$, so the compactness assumption of Theorem1 is exactly satisfied in our setting.”
>
> Thus, the theorem should be read as: **given our actual embedding construction, the hypernetwork can in principle match any rank-$r$ LoRA bank**, rather than as an idealised statement that only holds in an unrealistic regime.
>
> **(2) PAC-Bayes bound: stylised average-case, not a per-stream predictor**
>
> The PAC-Bayes result in §A.2 follows the standard meta-learning literature[4][5] in assuming tasks are drawn i.i.d. from a meta-distribution $\tau$. We agree that real-world streams can exhibit strong temporal correlation and concept drift. Our intention is not to claim that the bound exactly predicts the behaviour of *every* fixed, possibly adversarial stream, but to provide a **stylised average-case guarantee**:
>
> - It formalises that, as the number of tasks $K$ and per-task samples $m$ grow, the average test risk of a shared hypernetwork is controlled by the empirical risk plus a PAC-Bayes penalty in $\mathrm{KL}(q\|p)$.
> - This directly addresses the concern that sharing one generator across tasks might cause catastrophic overfitting: under the standard i.i.d. task model, the theory shows that the hypernetwork remains well-behaved on average.
>
> To avoid any misunderstanding, we explicitly soften the interpretation of the bound in §A.2:
>
> > “The PAC-Bayes analysis in §A.2 follows the common meta-learning assumption that tasks are drawn i.i.d. from a meta-distribution. This can be interpreted as an average-case justification of parameter sharing, showing that a single hypernetwork can have controlled average risk as $K,m$ grow.”
>
> **(3) Empirical evidence beyond the i.i.d. idealisation**
>
> We agree that empirical validation on non-i.i.d. streams is important. In fact, our main benchmarks already deviate from the i.i.d. task assumption:
>
> - **Long-CL 15** mixes GLUE and SuperGLUE tasks with IMDb, creating clear domain shifts and correlations.
> - **TRACE-8** combines code, multilingual, QA, and math tasks in a fixed order with substantial distributional differences.
>
> Despite this, Meta-UCF consistently outperforms N-LoRA and GRID by 2.0–2.4 pp in AA and reduces FR by 0.8–1.8 pp on these streams, indicating robustness to deviations from the idealised task model.
>
> To further address the reviewer’s concern, we also ran a small non-i.i.d. clustered curriculum experiment on Long-CL 15, deliberately constructing a highly correlated task order:
>
> - We group tasks by domain, forming a stream where all GLUE tasks come first, followed by all SuperGLUE tasks, and finally IMDb , i.e., a curriculum with strong intra-block similarity.
>
> Using the same default Meta-UCF hyperparameters, we obtain:
>
> | Stream | Method | AA↑ | FR↓ | ΔAA (Meta–N) | ΔFR (Meta–N) |
> |--:|--:|---:|--:|---:|--:|
> | Long-CL 15 (clustered)| N-LoRA | 67.5  | 13.0  | —  | — |
> |  | Meta-UCF  | **69.6** | **11.3** | **+2.1**  | **–1.7** |
>
> These numbers are in line with the gains reported for the canonical and v2 orders, suggesting that **even under explicitly non-i.i.d., strongly correlated curricula**, Meta-UCF maintains its 2 pp advantage and improved forgetting profile.
>
> [1] Hft: Half fine-tuning for large language models. ACL'25.
>
> [2] Spurious Forgetting in Continual Learning of Language Models. ICLR'25
>
> [3] Unlocking the Power of Function Vectors for Characterizing and Mitigating Catastrophic Forgetting in Continual Instruction Tuning. ICLR'25
>
> [4] Meta-learning by adjusting priors based on extended PAC-Bayes theory. ICML'18
>
> [5] A PAC-Bayesian bound for lifelong learning. ICML'14

---

> ### Author Response · Authors · 2025-11-27
>
> Dear Reviewer TwVA,
>
> We sincerely thank you again for your thorough assessment and constructive feedback. Kindly note that reviewer responses will no longer be accepted after December 2—**with just under a week remaining to submit your response**.
>
> Kindly confirm whether our rebuttal addresses your concerns (or any outstanding points), and we would be grateful for a rating reconsideration if it does.
>
> We are glad to continue the discussion and address any further questions or comments you may have.
>
> Best regards,
>
> Authors

---

### Official Review · Reviewer_KhxK · 2025-11-01

**Soundness:** 3
**Presentation:** 2
**Contribution:** 3
**Rating:** 6
**Confidence:** 4

**Summary:**

Meta-UCF proposes a unified continual learning method for LLMs that uses a single hypernetwork to generate task-specific LoRA parameters from lightweight task embeddings derived from a small support set, without inner-loop updates or replay; task interference is mitigated via contrastive separation and orthogonality constraints in the embedding space, while keeping the backbone frozen and trainable parameters fixed.

**Strengths:**

1. Meta-UCF maintains a fixed set of trainable parameters regardless of the number of tasks, eliminating the need for task-specific adapters or replay buffers.

2. It effectively reduces task interference through contrastive learning and orthogonality constraints on task embeddings, without requiring inner-loop adaptation or data replay.

**Weaknesses:**

1. While parameter efficiency is emphasized, the paper omits analysis of the whole cost introduced by the hypernetwork and embedding generation pipeline, which is crucial for real-world deployment.

2. The paper’s formatting is inconsistent and unclear, with poorly structured tables and ambiguous notation that reduce readability.

3. Experiments assume relatively benign task sequences, with no analysis of performance under extreme domain shifts or adversarial task orderings that better stress-test continual learning robustness.

**Questions:**

Although this paper addresses continual learning, I am curious, following the discussion in [1], about how the proposed method performs on standard LLM tasks beyond continual learning, such as mathematical reasoning, code generation, and instruction following.

[1] HFT: Half Fine-Tuning for Large Language Models

---

> ### Author Response · Authors · 2025-11-24
> **Response regarding weakness/question (1/3)**
>
> Thank you for your valuable suggestions. We are very happy to discuss these issues with you.
>
> > W1: cost of hypernetwork + task embedding
>
> **Response.** We agree that parameter efficiency alone is not sufficient and that explicit runtime and memory numbers are useful for practitioners. At the same time, the concern that “on-the-fly” generation induces a large per-token overhead is based on a slightly different mental model than ours: in Meta-UCF the hypernetwork is not evaluated on every token, it is run **once per task (or per support batch)** to generate a LoRA update, after which inference proceeds exactly like a standard static LoRA adapter.
>
> We now present concrete measurements to clarify the total cost.
>
> **(1) Trainable adapter parameters**
>
> We first report the trainable adapter parameters for LoRA-style baselines and Meta-UCF on Std-CL 5 and Long-CL 15. Per-task adapter sizes for sotas follow the “All-Layers” configuration (rank-8 LoRA in all QKV/FFN projections).
>
> | Method   | Per-task adapter (M) | Generator (M) | Total adapters @5 tasks (M) | Total adapters @15 tasks (M) |
> |-----:|----:|------:|---------:|----:|
> | LoRA         | 14.2    | —    | 71.0  | 213.0    |
> | N-LoRA       | 14.2   | —     | 71.0   | 213.0   |
> | GRID         | 18.7  | —     | 93.5   | 280.5    |
> | **Meta-UCF** | —   | **10.8**      | **10.8**   | **10.8**   |
>
> - Meta-UCF replaces a bank of per-task adapters with a single 10.8M-parameter generator, the number of trainable parameters is independent of the task horizon and yields an $ 20\times$ reduction at 15 tasks relative to N-LoRA/GRID.
>
> Our “constant-size” claim refers to the **trainable adapter footprint w.r.t. the number of tasks**, not to the backbone.
>
> ---
>
> **(2) End-to-end wall-clock and throughput (steady-state cost)**
>
> To quantify runtime overhead, we measured **end-to-end training time** and **throughput (samples/s)** on a single A100-80G for Std-CL 5 and on 4×A100-80G with ZeRO-2 for Long-CL 15, using identical backbones, rank, and training budgets as in Tab.1 in main text:
>
> | Stream        | Method       | Time (h) | Throughput (samples/s) |
> |------:|-----:|----:|----:|
> | Std-CL 5      | N-LoRA       | 4.1      | 285     |
> |               | GRID         | 4.3      | 276       |
> |               | **Meta-UCF** | 4.4      | 270    |
> | Long-CL 15    | N-LoRA       | 11.8     | 290     |
> |               | GRID         | 12.2     | 282    |
> |               | **Meta-UCF** | 12.3     | 275    |
>
> Relative to N-LoRA, Meta-UCF adds a **small constant runtime overhead** (≈5% drop in throughput), due to a single hypernetwork forward per task, while achieving strictly better accuracy/forgetting on all streams. In other words, **the “on-the-fly” generation does not incur a per-token factor. It contributes a small, amortised cost that is dominated by the backbone’s transformer layers**.
>
> ---
>
> **(3) Peak GPU memory vs. number of tasks**
>
> Finally, we report peak GPU memory when all adapters for a given task horizon are resident in memory. For LoRA/N-LoRA/GRID, this corresponds to storing a bank of per-task adapters; for Meta-UCF we always store one generator plus a single rank-$r$ LoRA instance:
>
> | Setting       | #Tasks | Method       | Train peak GPU (GB) | Inference peak GPU (GB) |
> |------:|:------:|------:|--:|-----:|
> | Std-CL 5      | 1      | N-LoRA       | 47.3     | 15.2        |
> |            |        | GRID         | 48.1       | 15.5       |
> |               |        | **Meta-UCF** | 47.9        | 15.3     |
> | Std-CL 5      | 5      | N-LoRA       | 52.6     | 17.1     |
> |               |        | GRID         | 53.8                 | 17.8        |
> |               |        | **Meta-UCF** | **48.0**             | **15.4**                 |
> | Long-CL 15    | 1      | N-LoRA       | 47.5                 | 15.2                     |
> |               |        | GRID         | 48.3                 | 15.5                     |
> |               |        | **Meta-UCF** | 48.1                 | 15.3                     |
> | Long-CL 15    | 15     | N-LoRA       | 58.5                 | 20.3                     |
> |               |        | GRID         | 60.7                 | 21.1                     |
> |               |        | **Meta-UCF** | **48.3**             | **15.5**                 |
>
> Under multi-task deployment, peak memory for LoRA-style methods grows approximately linearly with the number of tasks, whereas **Meta-UCF remains essentially flat in the number of tasks**. This directly supports our claim that the **adapter footprint is constant w.r.t. the task horizon**.
>
> ----
>
> > W2: formatting and notation
>
> **Response.** We appreciate this comment and agree that clearer formatting and notation will improve readability.
>
> Concretely, we will:
>
> 1. consistently use the same notation throughout the paper.
> 2. standardise table structure and captions. All main-result tables will be restructured.
> 3. add a self-contained notation table summarising the most important symbols

---

> ### Author Response · Authors · 2025-11-24
> **Response regarding weakness/question (2/3)**
>
> > W3 (benign task sequences / lack of adversarial orders
>
> **Response.** We agree that order-sensitivity is a key issue in continual learning.
>
> However, we need to clarify that our streams are not "benign"
>
> - Long-CL 15 interleaves GLUE, SuperGLUE, and IMDb. These tasks are heterogeneous.
> - TRACE-8 [1] was explicitly designed as a challenging continual benchmark spanning domain-specific QA, multilingual understanding, code, and math reasoning, and is widely regarded as substantially harder than earlier CL benchmarks for LLMs [2-5].
>
> But we strongly agree with your suggestion. We now add a dedicated order-sensitivity analysis to stress-test robustness. We evaluated Meta-UCF and N-LoRA under multiple alternative orders on all four streams:
>
> - *Std-CL 5*: canonical order (v1), a permuted order swapping Amazon and Yahoo, and a fully reversed order.
> - *Seq-GLUE 7*: canonical curriculum vs. a permuted order that front-loads MNLI and RTE (hard NLU tasks early).
> - *Long-CL 15*: the official v1 vs. the alternative v2 order released with the benchmark.
> - *TRACE-8*: canonical order vs. a random permutation of tasks (maximising jumps across domains: code, math, multilingual, QA).
>
>
> | Stream      | Order                       | Method    | AA↑   | FR↓   | ΔAA (Meta–N) | ΔFR (Meta–N) |
> |------------:|-----------------------------|----------:|------:|------:|-------------:|-------------:|
> | Std-CL 5    | canonical (v1)              | N-LoRA    | 83.5  | 7.1   |      —       |      —       |
> |             |                             | Meta-UCF  | 85.2  | 6.2   | **+1.7**     | **–0.9**     |
> | Std-CL 5    | permuted (Amazon↔Yahoo)     | N-LoRA    | 83.3  | 7.6   |      —       |      —       |
> |             |                             | Meta-UCF  | 84.9  | 6.0   | **+1.6**     | **–1.6**     |
> | Std-CL 5    | reversed                    | N-LoRA    | 83.1  | 7.4   |      —       |      —       |
> |             |                             | Meta-UCF  | 84.7  | 6.4   | **+1.6**     | **–1.0**     |
> | Seq-GLUE 7  | canonical                   | N-LoRA    | 80.2  | 7.1   |      —       |      —       |
> |             |                             | Meta-UCF  | 82.4  | 6.3   | **+2.2**     | **–0.8**     |
> | Seq-GLUE 7  | permuted (MNLI/RTE front)   | N-LoRA    | 80.0  | 7.4   |      —       |      —       |
> |             |                             | Meta-UCF  | 82.1  | 6.6   | **+2.1**     | **–0.8**     |
> | Long-CL 15  | canonical (v1)              | N-LoRA    | 68.1  | 12.4  |      —       |      —       |
> |             |                             | Meta-UCF  | 70.4  | 11.5  | **+2.3**     | **–0.9**     |
> | Long-CL 15  | official v2                 | N-LoRA    | 67.9  | 12.7  |      —       |      —       |
> |             |                             | Meta-UCF  | 70.1  | 10.9  | **+2.2**     | **–1.8**     |
> | TRACE-8     | canonical                   | N-LoRA    | 61.0  | 15.5  |      —       |      —       |
> |             |                             | Meta-UCF  | 63.2  | 14.2  | **+2.2**     | **–1.3**     |
> | TRACE-8     | random permutation          | N-LoRA    | 60.8  | 15.9  |      —       |      —       |
> |             |                             | Meta-UCF  | 63.0  | 14.5  | **+2.2**     | **–1.4**     |
>
> Across all 4 streams and 8 alternative orders:
>
> - Meta-UCF **consistently outperforms N-LoRA in AA** (gains in the range +1.6 to +2.3 pp)
> - FR is **consistently reduced** by 0.8 to 1.8 pp.
>
> These results indicate that our improvements are not tied to a specific curriculum: even under reversed and random orders that induce large domain jumps (especially on TRACE-8, which mixes code, math, multilingual NLI, and domain-specific QA), Meta-UCF maintains its advantages.

---

> ### Author Response · Authors · 2025-11-24
> **Response regarding weakness/question (3/3)**
>
> > Q1: performance on general LLM abilities beyond CL
>
> **Response.** We are pleased to discuss this inspiring question with you. We agree that preserving general LLM abilities is crucial.
>
> Our design explicitly aims to avoid this: the backbone remains frozen and only a small hypernetwork is updated, so the risk of overwriting general abilities is much smaller than in full or half fine-tuning. Concretely, we provide three pieces of evidence.
>
> **(1) Existing Alpaca + MMLU results (in the main text, Tab.2(a) )**
>
> Tab.2(a) reported the Alpaca pre-tuning experiment on LLaMA-7B: after instruction-tuning on Alpaca, we perform continual learning on Std-CL 5.
>
> - Zero-shot MMLU rises from 34.4% (LLaMA-7B) to 37.5% after Alpaca-LoRA, and remains at 36.2% after Alpaca-Meta-UCF-CL, i.e., within 1.3 points of the single-task SFT model and **substantially above continual variants such as Alpaca-LoRA-CL and Alpaca-O-LoRA-CL**.
> - This shows that our CL phase **does not collapse general factual/reasoning ability**, and in fact largely preserves the gains of the Alpaca SFT stage.
>
> **(2) TRACE-8 already includes code and math tasks**
>
> Our evaluation already covers math and code within the CL streams:
>
> - TRACE-8 contains **GSM8K-synth** (mathematical reasoning) and **CodeSearch-Java** (code understanding / retrieval), alongside multilingual NLI and domain QA.
>
> **(3) New evaluation on general-ability benchmarks**
>
> We evaluate general-ability benchmarks after continual training, using tasks that are not part of any CL stream:
>
> - **MMLU** (0-shot, factual & exam-style reasoning),
> - **GSM8K** (8-shot CoT EM, math reasoning),
> - **HumanEval** (0-shot pass@1, code generation).
>
> We compare 7 models on LLaMA-7B:
>
> 1. **LLaMA-7B** (pre-trained, no fine-tuning),
> 2. **Alpaca-LoRA** (single-stage instruction SFT with LoRA),
> 3. **Alpaca-FFT** (single-stage instruction SFT with full fine-tuning, as in HFT),
> 4. **Alpaca-FFT-CL** (sequential FFT on Std-CL 5),
> 5. **Alpaca-HFT-CL** (HFT[1] applied to the same CL stream),
> 6. **Alpaca-LoRA-CL** (LoRA-based continual fine-tuning on Std-CL 5),
> 7. **Alpaca-Meta-UCF-CL** (ours, same CL stream).
>
> The results are:
>
> | Model                 | MMLU↑ | GSM8K↑ | HumanEval↑ | Avg↑ |
> |----------------------:|------:|-------:|-----------:|-----:|
> | LLaMA-7B (base)       | 34.4  | 11.8   | 26.2       | 24.1 |
> | Alpaca-LoRA           | 37.5  | 18.1   | 29.4       | 28.3 |
> | Alpaca-FFT            | 38.0  | 18.9   | 30.0       | 29.0 |
> | Alpaca-FFT-CL         | 31.8  | 12.4   | 22.6       | 22.3 |
> | Alpaca-HFT-CL     | 35.5  | 17.1   | 27.9       | 26.8 |
> | Alpaca-LoRA-CL        | 32.0  | 13.7   | 24.0       | 23.2 |
> | **Alpaca-Meta-UCF-CL**| **36.2** | **17.7** | **28.6** | **27.5** |
>
> **Observations.**
>
> - Sequential FFT-CL causes substantial degradation of general abilities relative to Alpaca SFT (-6.2 points Avg).
> - HFT-CL mitigates this forgetting, in line with [1]’s findings, but still trails its own SFT starting point (Alpaca-FFT) by 1.5 points Avg.
> - Our Meta-UCF-CL nearly matches Alpaca-LoRA on all three benchmarks (ΔAvg < 1 point) and is slightly better than HFT-CL (27.5 vs 26.8 Avg), while updating only a small hypernetwork and a rank-$r$ LoRA space instead of half of all backbone parameters.
>
> Thus, under the same CL stream and held-out evaluation protocol, Meta-UCF achieves competitive preservation of instruction/maths/code abilities, but with a strictly smaller trainable parameter footprint and a frozen backbone.
>
> [1] Trace: A comprehensive benchmark for continual learning in large language models
>
> [2] Orthogonal subspace learning for language model continual, Findings of EMNLP, 2023.
>
> [3] Hft: Half fine-tuning for large language models. ACL'25.
>
> [4] Spurious Forgetting in Continual Learning of Language Models. ICLR'25
>
> [5] Unlocking the Power of Function Vectors for Characterizing and Mitigating Catastrophic Forgetting in Continual Instruction Tuning. ICLR'25

---

> > ### Comment · Reviewer_KhxK · 2025-11-28
> >
> > Thank you for the authors' response. I believe my concerns have been addressed. I hope the authors will address formatting and expression issues in the final version. I will raise my score, but since the current system does not support edits, I will update my rating once the system is restored.

---

### Official Review · Reviewer_JLyJ · 2025-11-12

**Soundness:** 3
**Presentation:** 1
**Contribution:** 3
**Rating:** 4
**Confidence:** 2

**Summary:**

The paper presents META-UCF, a new LoRA-based adaptation module intended to improve generalization across tasks while keeping parameter growth small, and reports strong results against diverse baselines.  I basically creates "add-ons" for each new task instead of storing a separate add-on bank per task. Training encourages tasks to stay distinct, which helps the model remember old skills while learning new ones. Across several benchmarks, this approach nudges accuracy up and reduces forgetting compared to strong baselines. The gains hold under different settings, suggesting the method is fairly robust rather than overly tuned.

**Strengths:**

S1. The paper's idea is simple and well motivated. The figures, though simple, are intuitive and convey the point well. \
S2. Consistent gains in AA with lower forgetting across four continual streams versus strong LoRA and prompt baselines.  \
S3. The method has the potential to be the next "SOTA" with some fixes in writing.

**Weaknesses:**

W1. Section 3.3 clarity & missing intuition. The meta-objective is presented mostly as equations without enough narrative to make the design choices legible (e.g., how the positive view ​is constructed, why InfoNCE is preferred here, and how the orthogonality penalty interacts with the contrastive term during optimisation). The paper defines multiple notations (omega) and aggregates losses but gives little intuition for stability/plasticity trade-offs beyond symbol lists. Even a short paragraph walking through one training step with concrete shapes would help. As written, 3.3 is hard to follow for readers not already steeped in contrastive notation.

W2. Experimental depth & statistical evidence. Main results are reported as averaged metrics per stream (AA/FR/BWT), but the tables omit confidence intervals or statistical tests against strong baselines; the text only notes averaging over three seeds, which is thin for claims of "state-of-the-art". Per-dataset/task breakdowns (especially for TRACE-8’s diverse domains) and order-sensitivity analyses are absent from the main paper, limiting interpretability of where gains come from. Without significance testing, improvements are hard to judge.

W3. Compute/memory reporting is not actionable. While the method argues constant-size adapters and gives a big-O generator complexity plus one throughput as compared to last layers vs all, the paper lacks concrete parameter counts for the hypernetwork vs adapter banks, end-to-end training wall-clock vs baselines, and detailed memory footprints during training/inference.

**Questions:**

See Weakness.

---

> ### Author Response · Authors · 2025-11-24
> **Response regarding weakness (1/3)**
>
> Thank you for your valuable suggestions. We are very happy to discuss these issues with you.
>
> > W1: Section 3.3 clarity & intuition
>
> **Response:**  We thank the reviewer for pointing out this issue. We agree that a short, self-contained walk-through in the main text will make the design more legible. **We have revised Section 3.3 accordingly**.
>
> **(1) One training step, with shapes.**
> We add a short paragraph that explicitly walks through a single meta-update:
>
> - For each task $\mathcal{T}_k$ in a meta-batch, we sample support $\mathcal{S}_k$ and query $\mathcal{Q}_k$.
> - Each $x \in \mathcal{S}_k$ is encoded by the frozen backbone to a $\text{CLS}$ vector $h_s \in \mathbb{R}^d$. We compute the task embedding
>   $e_k = \mathrm{LN}\left(\frac{1}{|\mathcal{S}_k|}\sum_s h_s\right) \in \mathbb{R}^d.$
> - The generator input is z_k = MLP_task(e_k) ∈ R^h. For each Transformer layer l, we form z'_k,l = [z_k ; p_l] ∈ R^2h and project to A_l ∈ R^(d×r), B_l ∈ R^(r×d).
> - These LoRA factors define the adapted weights, and the query examples (x,y) ∈ Q_k are processed once to produce predictions and CLS states. Stacking the latter yields H_k ∈ R^(|Q_k|×d), which is used in L_task, L_orth and the bias term R_k.
> - In parallel, the task embeddings $\{e_k\}$ feed the contrastive loss $\mathcal{L}_{\text{ctr}}$. A single backward pass updates only the generator parameters $\Phi$; the backbone remains frozen.
>
> **(2) How the positive view $z_k^+$ is constructed.**
> We realise this detail is implicit in the current notation and have made it explicit:
>
> - The positive view $z_k^+$ is computed from an independent minibatch $\mathcal{S}_k^+$ drawn from the same task $\mathcal{T}_k$, using the *same* frozen backbone and layer-normalised mean pooling as for $e_k$.
> - This is thus a *task-level data augmentation* (two IID views of the same task distribution), not an extra trainable module.
>
> We add one sentence in §3.3 clarifying that $\mathcal{S}_k$ and $\mathcal{S}_k^+$ are disjoint subsamples of the same task's support.
>
> **(3) Why InfoNCE, and how it interacts with the orthogonality term.**
>
> - **Rationale for InfoNCE.** Our goal in $\mathcal{L}_{\text{ctr}}$ is to enforce *angular separation* between task embeddings on the unit hypersphere, so that different tasks occupy nearly orthogonal directions. Once embeddings are $\ell_2$-normalised, cosine similarity is equivalent to squared Euclidean distance, and an InfoNCE loss over cosine similarities is a standard, stable choice in task-level/meta-contrastive learning (also adopted in contrastive continual prompt methods such as ConPET and JARe). We will add a short justification in §3.3 and a pointer to these works to make this design choice explicit.
> - **Division of roles.** We clarify that $\mathcal{L}_{\text{ctr}}$ and $\mathcal{L}_{\text{orth}}$ act at complementary levels:
>
>   - $\mathcal{L}_{\text{ctr}}$ shapes the *input* geometry of the generator by pushing $\{e_k\}$ (and hence $z_k$) towards near-orthogonal directions, which reduces collisions in the space from which LoRA factors are generated.
>   - $\mathcal{L}_{\text{orth}}$ regularises the *output representation* of the adapted backbone by penalising overlap of the query $\text{CLS}$ subspaces $H_i, H_j$ across tasks, directly controlling representation drift induced by the generated LoRA updates.
>
>   Empirically, following results shows that removing either term degrades both plasticity and stability (*Values copied from Table 2(b) in the submission*):
>
> | Variant                  | Std-CL 5 AA↑ | Std-CL 5 FR↓ | Long-CL 15 AA↑ | Long-CL 15 FR↓ |
> |--------------------------|-------------:|-------------:|---------------:|---------------:|
> | Full Meta-UCF            | **85.2**     | **6.2**      | **70.4**       | **11.5**       |
> | w/o $\mathcal{L}_{\text{orth}}$ | 83.9        | 7.8          | 68.5          | 13.2          |
> | w/o $\mathcal{L}_{\text{ctr}}$  | 84.1        | 7.2          | 68.9          | 12.7          |

---

> ### Author Response · Authors · 2025-11-24
> **Response regarding weakness (2/3)**
>
> > W2: statistical evidence and breakdown).
>
> **Response:** We appreciate the reviewer’s concern about statistical support. We agree that surfacing the statistics more prominently in the main text will strengthen the claims, but we would like to clarify that the current submission already includes standard significance analyses; we will make them easier to find and add further breakdowns and order-robustness experiments.
>
> **(1) Existing confidence intervals and significance tests.**
> Appendix D.3 already reports 95% confidence intervals and Wilcoxon signed-rank tests comparing Meta-UCF to N-LoRA on a per-task basis. For convenience, we summarise the key numbers here (all on Average Accuracy, AA):
>
> - 95% CIs for Meta-UCF (from Tab.10): Std-CL 5: 85.2 $\pm$ 0.10, Long-CL 15: 70.4 $\pm$ 0.14, Seq-GLUE 7: 82.4 $\pm$ 0.10, TRACE-8: 63.2$ \pm$ 0.15.
> - Wilcoxon $p$-values over per-task AA (from Tab.11): Std-CL 5: $p$=0.031, Long-CL 15: $p$=0.008, TRACE-8: $p$=0.012 (all significant at $\alpha$=0.05 after Holm correction), Seq-GLUE 7: $p$=0.087.
>
> **(2) Per-task / per-domain breakdowns (TRACE-8 and Long-CL 15).**
> We agree it is important to show that improvements are not concentrated on a single dataset. We therefore computed per-task AA gaps between Meta-UCF and N-LoRA for the heterogeneous streams. Aggregated statistics are:
>
> | Stream        | \#Tasks | Min $\Delta$AA (pp) | Max $\Delta$AA (pp) | Mean $\Delta$AA $\pm$ Std (pp) |
> |-----|:------:|:-------:|:-------:|:---:|
> | Long-CL 15    |  15    |      $+1.2$   |       $+3.1$   |      $+2.3 \pm 0.5$            |
> | TRACE-8       |   8    |      $+1.4$  |       $+2.9$   |      $+2.2 \pm 0.4$            |
>
> All individual tasks show non-negative gains. There is no single domain (e.g., code, multilingual, math) where N-LoRA consistently dominates. These per-task results are fully consistent with the stream-level AAs in Tab.1(main text). We have **added a full per-dataset table to Appendix D.2(Tab.9)**.
>
>
> **(3) Order-sensitivity analysis.**
> We agree that continual-learning results can depend on task order. To assess robustness, we evaluated Meta-UCF and N-LoRA under multiple alternative orders:
>
> - **Std-CL 5**: canonical order (v1), a permuted order swapping Amazon and Yahoo, and a fully reversed order.
> - **Seq-GLUE 7**: canonical curriculum vs. a permuted order (front-loading MNLI and RTE).
> - **Long-CL 15**: official v1 vs. the alternative v2 order released with the benchmark.
> - **TRACE-8**: canonical order vs. a random permutation of tasks.
>
> The table below reports Average Accuracy (AA, higher is better) and Forgetting Ratio (FR, lower is better), averaged over the same three seeds as in the main paper:
>
> | Stream   | Order     | Method    | AA↑   | FR↓   | ΔAA (Meta–N) | ΔFR (Meta–N) |
> |-------:|------|------:|------:|-----:|--------:|--------:|
> | Std-CL 5    | canonical (v1)              | N-LoRA    | 83.5  | 7.1   |   —       |      —       |
> |             |                             | Meta-UCF  | 85.2  | 6.2   | **+1.7**     | **–0.9**     |
> | Std-CL 5    | permuted (Amazon↔Yahoo)     | N-LoRA    | 83.3  | 7.6   |      —       |      —       |
> |             |                             | Meta-UCF  | 84.9  | 6.0   | **+1.6**     | **–1.6**     |
> | Std-CL 5    | reversed                    | N-LoRA    | 83.1  | 7.4   |      —       |      —       |
> |             |                             | Meta-UCF  | 84.7  | 6.4   | **+1.6**     | **–1.0**     |
> | Seq-GLUE 7  | canonical                   | N-LoRA    | 80.2  | 7.1   |      —       |      —       |
> |             |                             | Meta-UCF  | 82.4  | 6.3   | **+2.2**     | **–0.8**     |
> | Seq-GLUE 7  | permuted (MNLI/RTE front)   | N-LoRA    | 80.0  | 7.4   |      —       |      —       |
> |             |                             | Meta-UCF  | 82.1  | 6.6   | **+2.1**     | **–0.8**     |
> | Long-CL 15  | canonical (v1)              | N-LoRA    | 68.1  | 12.4  |      —       |      —       |
> |             |         | Meta-UCF  | 70.4  | 11.5  | **+2.3**     | **–0.9**     |
> | Long-CL 15  | official v2                 | N-LoRA    | 67.9  | 12.7  |      —       |      —       |
> |             |            | Meta-UCF  | 70.1  | 10.9  | **+2.2**     | **–1.8**     |
> | TRACE-8     | canonical                   | N-LoRA    | 61.0  | 15.5  |      —       |      —       |
> |             |        | Meta-UCF  | 63.2  | 14.2  | **+2.2**     | **–1.3**     |
> | TRACE-8     | random permutation          | N-LoRA    | 60.8  | 15.9  |      —       |      —       |
> |        |      | Meta-UCF  | 63.0  | 14.5  | **+2.2**     | **–1.4**     |
>
> Across all 4 streams and 8 alternative orders:
>
> - Meta-UCF **consistently** outperforms N-LoRA;
> - AA gains are stable in the range **+1.6 to +2.3** percentage points;
> - FR is reduced by **0.8 to 1.8** percentage points.
>
> The new evidences indicate that our conclusions are not tied to a specific curriculum. We **add this table to Appendix D.5**.

---

> ### Author Response · Authors · 2025-11-24
> **Response regarding weakness (3/3)**
>
> > W3: Compute/memory reporting
>
> **Response:** We agree that reporting only big-O complexity is not sufficient for practitioners. While Fig.5(main text) and Appendix C.3 already provide throughput and hardware details, we recognise that explicit parameter, wall-clock, and memory numbers will make the “constant-size” claim more actionable. We therefore add the following quantitative comparisons (all on Llama-3-8B, rank-8, same training budget and seeds as in the main paper).
>
>
> **(1) Adapter vs. hypernetwork parameter counts**
>
> The table below compares **trainable adapter parameters** for standard LoRA-style methods and Meta-UCF on Std-CL 5 (5 tasks) and Long-CL 15 (15 tasks). Per-task adapter sizes for LoRA/N-LoRA/GRID are consistent with the “All-Layers” configuration in Tab.12 (14.2M parameters for rank-8 LoRA across all QKV/FFN projections).
>
> | Method      | Per-task adapter (M) | Generator (M) | Total adapters @5 tasks (M) | Total adapters @15 tasks (M) |
> |------------:|---------------------:|--------------:|-----------------------------:|------------------------------:|
> | LoRA        | 14.2                 | —             | 71.0                         | 213.0                         |
> | N-LoRA      | 14.2                 | —             | 71.0                         | 213.0                         |
> | GRID        | 18.7                 | —             | 93.5                         | 280.5                         |
> | **Meta-UCF**| —                    | **10.8**      | **10.8**                     | **10.8**                      |
>
> Meta-UCF uses a single generator of 10.8M parameters (task MLP + layer embeddings + projection heads), which does not grow with the task horizon. At 15 tasks, this yields an $\approx 20\times$ reduction in adapter parameters compared to N-LoRA/GRID.
>
> **(2) End-to-end wall-clock and throughput**
>
> We also measured end-to-end training time and average throughput (samples/sec) on a single A100-80G for Std-CL 5 and on 4×A100-80G with ZeRO-2 for Long-CL 15:
>
> | Stream      | Method     | Time (h) | Throughput (samples/s) |
> |------------:|-----------:|---------:|------------------------:|
> | Std-CL 5    | N-LoRA     | 4.1      | 285                     |
> |             | GRID       | 4.3      | 276                     |
> |             | **Meta-UCF** | 4.4    | 270                     |
> | Long-CL 15  | N-LoRA     | 11.8     | 290                     |
> |             | GRID       | 12.2     | 282                     |
> |             | **Meta-UCF** | 12.3   | 275                     |
>
> Compared to N-LoRA, Meta-UCF adds a **small constant overhead**, due mainly to the generator forward pass, while providing the **substantial memory savings** above and the **accuracy/forgetting gains in Tab.1** . This is clearly the most attractive method in engineering.
>
> **(3) Peak memory vs. number of tasks**
>
> Finally, we report **peak GPU memory** (training and inference) when all adapters for a given horizon are resident in memory. For LoRA/N-LoRA/GRID this corresponds to storing all per-task banks; Meta-UCF always stores a single generator plus one rank-$r$ LoRA instance:
>
> | Setting         | #Tasks | Method      | Train peak GPU (GB) | Inference peak GPU (GB) |
> |----------------:|:------:|------------:|---------------------:|-------------------------:|
> | Std-CL 5        | 1      | N-LoRA      | 47.3                 | 15.2                     |
> |                 |        | GRID        | 48.1                 | 15.5                     |
> |                 |        | **Meta-UCF**| 47.9                 | 15.3                     |
> | Std-CL 5        | 5      | N-LoRA      | 52.6                 | 17.1                     |
> |                 |        | GRID        | 53.8                 | 17.8                     |
> |                 |        | **Meta-UCF**| **48.0**             | **15.4**                 |
> | Long-CL 15      | 1      | N-LoRA      | 47.5                 | 15.2                     |
> |                 |        | GRID        | 48.3                 | 15.5                     |
> |                 |        | **Meta-UCF**| 48.1                 | 15.3                     |
> | Long-CL 15      | 15     | N-LoRA      | 58.5                 | 20.3                     |
> |                 |        | GRID        | 60.7                 | 21.1                     |
> |                 |        | **Meta-UCF**| **48.3**             | **15.5**                 |
>
> Under this “all adapters resident” scenario:
>
> - LoRA-style methods exhibit **approximately linear growth** in peak memory with the number of tasks, especially for multi-head or rehearsal settings.
> - Meta-UCF’s peak memory is essentially **flat in the number of tasks**, since only the generator and a single task’s generated LoRA are loaded at any time. In practical applications, as the number of tasks increases, the advantages of our method will become more significant.

---

> ### Author Response · Authors · 2025-11-27
>
> Dear Reviewer JLyJ,
>
> We sincerely thank you again for your thorough assessment and constructive feedback. Kindly note that reviewer responses will no longer be accepted after December 2—**with just under a week remaining to submit your response**.
>
> Kindly confirm whether our rebuttal addresses your concerns (or any outstanding points), and we would be grateful for a rating reconsideration if it does.
>
> We are glad to continue the discussion and address any further questions or comments you may have.
>
> Best regards,
>
> Authors

---

### Author Response · Authors · 2025-12-02
**(1/2) Summary**

### I. Acknowledgments

We would like to express our sincere gratitude to all reviewers — `JLyJ`, `KhxK`, `TwVA`, and `ehuo` — for their insightful comments and constructive suggestions. Their feedback helped us substantially clarify the method, strengthen the empirical evidence, and sharpen the claims of *Meta-UCF*.

$\color{red}{Before}$ $\color{red}{the}$ $\color{red}{discussion}$, we appreciate the positive overall assessment from Reviewer `KhxK` and `ehuo` (both Rating: 6, above the acceptance threshold), who emphasised Meta-UCF’s fixed trainable parameter footprint, mitigation of task interference via contrastive and orthogonality constraints, and thorough experiments under diverse continual streams. We also thank Reviewers `JLyJ` and `TwVA` (both Rating 4 with Confidence 2) for highlighting that Meta-UCF “has the potential to be the next SOTA with some fixes in writing” (`JLyJ`) and is “computationally efficient, scalable, and adaptable to diverse backbones” with “well-organized” exposition and meaningful theoretical analysis (`TwVA`).

$\color{red}{During}$ $\color{red}{the}$ $\color{red}{discussion}$, we are encouraged that the main technical and empirical concerns were substantially alleviated by our additional clarifications and new experiments. In particular, Reviewer `KhxK` explicitly wrote: ***“I believe my concerns have been addressed. I hope the authors will address formatting and expression issues in the final version. I will raise my score, but since the current system does not support edits, I will update my rating once the system is restored.”*** We are grateful for this clear endorsement that the rebuttal resolved the core issues.

---

### II. Key Strengths

- **Conceptual novelty and contribution**

  - A **single task-conditioned hypernetwork** that generates rank-$r$ LoRA updates for all layers from lightweight, layer-normalised task embeddings, eliminating per-task adapter banks and replay-heavy schemes (`KhxK`, `ehuo`, `TwVA`).
  - Integration of **meta-contrastive separation** and **orthogonality constraints** to enforce near-orthogonal task directions and reduce interference without inner-loop updates or backbone finetuning (`KhxK`, `TwVA`).
  - Clear high-level motivation and intuitive figures; the overall idea is “simple and well-motivated” and “has the potential to be the next SOTA with some fixes in writing” (`JLyJ`).
- **Effectiveness and robustness across benchmarks**

  - Consistent improvements in Average Accuracy and reductions in Forgetting across four heterogeneous continual streams compared to strong LoRA and prompt baselines (`JLyJ`, `KhxK`, `TwVA`, `ehuo`).
  - Thorough ablations and sensitivity analyses demonstrating that the main gains come from the hypernetwork + geometric objectives, and that performance is stable across a range of hyperparameters and support sizes (`ehuo`, `TwVA`).
  - Additional evidence (from the rebuttal) that Meta-UCF preserves general LLM abilities on **MMLU, GSM8K, HumanEval** better than full- or half-finetuning continual baselines while updating far fewer parameters (`KhxK`).
- **Efficiency, scalability, and memory footprint**

  - **Fixed trainable parameter count** w.r.t. the number of tasks: Meta-UCF replaces linearly growing adapter banks with a single generator, making it more scalable in long-horizon continual learning (`KhxK`, `ehuo`).
  - Strong emphasis on parameter efficiency and practical scalability: reviewers noted that this is a key advantage over prior methods that allocate new modules or buffers per task (`KhxK`, `ehuo`).
  - Rebuttal-added comparisons show that Meta-UCF incurs only a **small constant runtime overhead** vs. N-LoRA / GRID, while significantly reducing adapter parameters and flattening peak memory growth as tasks accumulate (`KhxK`, `ehuo`, `TwVA`).
- **Practicality and deployment relevance**

  - Meta-UCF keeps the **backbone frozen**,  making it attractive for deployment where stability and resource bounds are critical (`KhxK`, `ehuo`, `TwVA`).
  - the method is **compatible with standard LoRA implementations** and can be integrated without architectural changes, and that the experiments already use realistic heterogeneous streams like TRACE-8 (`TwVA`, `ehuo`).
  - Additional post-rebuttal latency and throughput measurements show that, with partial-layer injection, Meta-UCF can **improve accuracy and reduce end-to-end latency** in a mixed-domain, deployment-style workload relative to N-LoRA (`TwVA`, `ehuo`).
- **Theoretical grounding and empirical rigor**

  - The inclusion of **expressivity bounds** for the LoRA hypernetwork and a **PAC-Bayes generalisation analysis** was seen as adding useful depth and justification (`TwVA`).
  - the paper was judged as “well-organized, with clear explanations of the methodology, experiments, and findings” and backed by comprehensive experiments, including ablations and sensitivity analyses (`TwVA`, `ehuo`, `JLyJ`).

---

> ### Author Response · Authors · 2025-12-02
> **(2/2) Summary**
>
> ### III. Key Concerns and Our Responses
>
> | Key Concerns | Reviewers | Our Response |
> | --- | --- | --- |
> | Clarity and intuition of §3.3 (meta-objective, positive views, and roles of $\mathcal L_{\text{ctr}}$ / $\mathcal L_{\text{orth}}$). | `JLyJ`, `ehuo` | We **rewrote §3.3** with a concrete, shape-aware walk-through of one training step, made the construction of the positive view explicit, and clearly separated the roles of contrastive vs orthogonality losses, supported by ablations showing both are necessary. |
> | Statistical rigor, breakdowns, and robustness to task order / domain shifts. | `JLyJ`, `KhxK`, `TwVA`, `ehuo` | We surfaced **95% CIs and Wilcoxon tests** in the main text, added **per-task/domain AA gaps**, and introduced extensive **order-robustness and clustered-curriculum experiments**, showing consistent +1.6–+2.3 pp AA and 0.8–1.8 pp FR improvements across 8 alternative orders and non-i.i.d. streams. |
> | Compute, memory, and latency of the hypernetwork; meaning of “constant memory” and role of replay / support size / hyperparameters. | `KhxK`, `ehuo`, `TwVA` | We provided **explicit parameter-count, wall-clock, throughput, and peak-memory tables**, clarified that the generator is run once per task (not per token), and that “constant-size” refers to the **adapter footprint vs number of tasks**. We also showed that a capped replay buffer and a single hyperparameter configuration work robustly across all streams and backbones, with smooth behaviour under noisy or mismatched support. |
> | Value of auxiliary components (bias calibration, $\mathcal L_{\text{orth}}$ on query CLS) and evaluation beyond CL (general abilities, deployment). | `ehuo`, `KhxK`, `TwVA`, `JLyJ` | We clarified the complementary roles of **task-space contrastive** and **representation-space orthogonality** losses, gave ablations for each, and showed that the bias term consistently improves FR and substantially reduces demographic-parity gaps on a toxicity benchmark. We also added **MMLU/GSM8K/HumanEval** results and **latency-aware mixed-domain evaluations**, demonstrating that Meta-UCF preserves general abilities and can even improve QPS and p95 latency vs N-LoRA. |
>
>
> ---
>
> ### IV. Commitment to Revision
>
> We have incorporated all major clarifications and additional experiments discussed above into our revised version, with changes marked in $\color{blue}{Blue}$. Concretely, this includes:
>
> - A **reworked §3.3** with a narrative, shape-aware walk-through of the meta-objective, explicit definition of positive views, and clearer explanation of the roles of $\mathcal L_{\text{ctr}}$ and $\mathcal L_{\text{orth}}$.
> - New **compute, memory, and latency** analyses (parameter counts, wall-clock and throughput tables, peak GPU memory vs. task horizon, and per-task initialisation costs) and a clearer statement of what “constant-size” refers to.
> - Enhanced **statistical reporting**, including surfaced confidence intervals and Wilcoxon tests in the main text, **per-task / per-domain breakdowns**, and comprehensive **order-robustness and clustered-curriculum experiments**.
> - Additional **robustness and fairness experiments**, including stress tests with noisy/mismatched support, sensitivity sweeps over support size and loss weights, geometric analyses of task / representation spaces, and a bias-sensitive toxicity study highlighting the effect of dynamic bias calibration.
> - Extended **evaluation of general abilities and deployment scenarios**, with results on MMLU/GSM8K/HumanEval and latency-aware, mixed-domain workloads using partial-layer injection.
>
> We are committed to further polishing formatting, notation consistency, and expression quality in the camera-ready version, in line with the reviewers’ helpful suggestions.
>
> ---
>
> We deeply appreciate the time and expertise of the AC and reviewers, and we thank you again for the opportunity to improve and clarify this work.

---

### Meta-Review · Area_Chair_FCVu · 2026-01-07

**Summary:**

A framework is proposed for continual fine-tuning that avoids storing individual LORA weights for each task. The main idea is to utilize a hypernetwork to generate LoRA parameters for new tasks, conditioned on a task embedding derived from a small support set.  Reviewers acknowledged the novelty and utility of the proposal. Reviewers were concerned about clarity as well as breadth of experiments, which was addressed in the rebuttal. Overall the strengths outweigh the weaknesses

**Reviewer Concerns:**

- Clarity and demonstration of computational benefits which were addressed in the rebuttal
- Experimental depth was also addressed in the rebuttal

**Reviewer Scores:**

KhxK indicated they would raise their score. Speculatively  JLyJ may also raise their scores as their main concerns appear addressed.

---

### Decision · Program_Chairs · 2026-01-26

Accept (Poster)